# A General Analysis of Example-Selection for Stochastic Gradient Descent

**Yucheng Lu**[*]**, Si Yi Meng**[*]**, Christopher De Sa**
Department of Computer Science
Cornell University
Ithaca, NY 14853, USA
{yl2967,sm2833,cmd353}@cornell.edu

## Abstract

Training example order in SGD has long been known to affect convergence rate. Recent results show that accelerated rates are possible in a variety of cases for permutation-based sample orders, in which each example from the training set is used once before any example is reused. In this paper, we develop a broad condition on the sequence of examples used by SGD that is sufficient to prove tight convergence rates in both strongly convex and non-convex settings. We show that our approach suffices to recover, and in some cases improve upon, previous state-of-the-art analyses for four known example-selection schemes: (1) shuffle once, (2) random reshuffling, (3) random reshuffling with data echoing, and (4) Markov Chain Gradient Descent. Motivated by our theory, we propose two new example-selection approaches. First, using quasi-Monte-Carlo methods, we achieve unprecedented accelerated convergence rates for learning with data augmentation. Second, we greedily choose a fixed scan-order to minimize the metric used in our condition and show that we can obtain more accurate solutions from the same number of epochs of SGD. We conclude by empirically demonstrating the utility of our approach for both convex linear-model and deep learning tasks. Our code is available at: https://github.com/EugeneLYC/qmc-ordering.

## 1 Introduction

To minimize a differentiable function $f : \mathbb{R}^d \to \mathbb{R}$, stochastic gradient descent (SGD) iteratively updates a parameter vector $w \in \mathbb{R}^d$ starting at some $w_0$ by running

$$w_{t+1} = w_t - \alpha_t \nabla f(w_t; x_t), \qquad (1)$$

where $\alpha_t$ is the step size at iteration $t$, $x_t$ is a data *example* (often a minibatch of data) chosen by some process—typically by subsampling a training dataset—for SGD to use at iteration $t$, and $\nabla f(w_t; x_t)$ is an *example gradient*, which we hope will be a good approximation for the gradient of the objective $\nabla f(w_t)$. In the standard setup, the $x$'s are drawn from a dataset $\mathcal{D}$ of size $n$, and $f(w) = \frac{1}{n} \sum_{x \in \mathcal{D}} f(w; x)$, which is often referred to as Empirical Risk Minimization (ERM). The order in which the example sequence $x_0, x_1, \dots$ are chosen is known to affect the convergence of SGD. For instance, compare so-called "random reshuffling" to with-replacement sampling: optimizing a strongly convex objective with $T$ total iterations, random reshuffling samples the $x_t$ from $\mathcal{D}$ without replacement and achieves an *accelerated* convergence rate of $1/T^2$, while with-replacement sampling yields a convergence rate of $1/T$ (Bottou, 2012; Recht & Ré, 2012; Gürbüzbalaban et al., 2021). Similar accelerated rates have been shown in other settings and for other example orders, such as shuffling the dataset once (Nguyen et al., 2020; Ahn et al., 2020; Mishchenko et al., 2020). However, these analyses have mostly focused only on specific example-selection schemes, and study only the case of ERM-type finite-sum objectives. This does not help us understand how new example orders (such as the data echoing method of Choi et al. (2019)) affect the convergence of SGD.

This paper develops a general condition on the example gradients themselves that is sufficient to provide a convergence rate for SGD. Intuitively, our main result is: *the convergence rate of SGD*

---

[*]Equal Contribution.

*depends on how fast the averages of consecutive example gradients $\nabla f(w; x_t)$ converge to the full objective gradient $\nabla f(w)$.* Using with-replacement sampling, the average of $m$ consecutive gradient examples starting at any timestep $\tau$, $\frac{1}{m} \sum_{t=\tau}^{\tau+m-1} \nabla f(w; x_t)$, converges to $\nabla f(w)$ at a rate of $\mathcal{O}(1/m)$ in terms of the norm squared: we show SGD with any example sequence that fulfills this condition will converge at the same asymptotic rate as with-replacement sampling. Alternatively, if that average converges at the faster $\tilde{\mathcal{O}}(1/m^2)$ rate typical of *Quasi-Monte-Carlo* (QMC) (Caflisch, 1998), then we show SGD enjoys the accelerated rate that random reshuffling gets. Our contributions are as follows:

- We propose a new condition on the example gradients—*average gradient error*—and provide convergence analysis using this general condition for both non-convex and strongly-convex problems. We justify the validity of this condition on synthetic experiments (Section 3).
- We show that many commonly used example orderings—shuffle once, random reshuffling, data echoing, Markov Chain Gradient Descent—can be analyzed as special cases under our theoretical results, which match or improve upon their existing rates in the literature (Section 4).
- We propose two new algorithms: (1) QMC-based data augmentation that transforms examples via a low-discrepancy sequence, improving generalization; and (2) a greedy algorithm that sorts the examples before each epoch based on our new average gradient error metric (Section 5).
- Empirically, we evaluate our two algorithms on several image classification benchmarks including MNIST, CIFAR10/100 and ImageNet. We show with QMC-based data augmentation, a higher validation accuracy can be achieved without hyperparameter tuning—this suggests that QMC may be a good default driver to use with data augmentation for deep learning in general. Meanwhile, the greedy algorithm converges faster both in terms of iteration and wall-clock time (Section 6).

## 2 RELATED WORK

**Example ordering in stochastic optimization.** Traditional example ordering in SGD is carried out in a with-replacement fashion, which is used to ensure unbiased estimation of the full gradient (Robbins & Monro, 1951; Bach & Moulines, 2011; Zhang, 2004; Bottou et al., 2018; Drori & Shamir, 2020). Significant attention has been paid to importance sampling with respect to various measures, such as Lipschitz constants (Schmidt et al., 2017; Needell et al., 2014), example gradient norms and bounds (Zhao & Zhang, 2015; Alain et al., 2015; Papa et al., 2015; Lee et al., 2019), individual losses (Kawaguchi & Lu, 2020; Loshchilov & Hutter, 2015), and data heterogeneity (Lu et al., 2021). Without-replacement sampling, however, is more common in practice and empirically allows faster convergence (Bottou, 2012). Among the most popular without-replacement approaches are shuffle once (SO) (Bertsekas, 2011; Gürbüzbalaban et al., 2019) and random reshuffling (RR) (Ying et al., 2017). In theory, Recht & Ré (2012) undertook the first investigation on convergence of RR via the noncommutative arithmetic-geometric mean conjecture, to which subsequent works provide counter examples (Yun et al., 2021; De Sa, 2020). HaoChen & Sra (2019) performed an epoch-wise acceleration analysis on RR while Gürbüzbalaban et al. (2021) considered its convergence over infinite epochs. Safran & Shamir (2020) analyzed the lower bounds for both SO and RR methods, and their results are further polished by Mishchenko et al. (2020) via the Bregman divergence bound. Aside from manual ordering, another line of research focuses on perturbed example ordering from data echoing (Choi et al., 2019; Agarwal et al., 2020).

**Quasi-Monte Carlo.** Quasi-Monte Carlo (QMC) is a variant of Monte Carlo (MC) methods that uses a low-discrepancy sequence instead of a pseudorandom sequence. QMC has been successfully applied in a wide variety of domains including computer graphics (Keller, 1995), finance (Joy et al., 1996), and computational biology (Cieslak et al., 2008). In machine learning, using QMC in place of MC can significantly improve many techniques including variational inference (Buchholz et al., 2018; Liu & Owen, 2021), feature mapping (Yang et al., 2014; Avron et al., 2016), normalizing flows (Wenzel et al., 2018), deep learning based PDE (Chen et al., 2019), and time series analysis (Philipson et al., 2020). For stochastic optimization, early works like Homem-de Mello (2008) and Pennanen (2005) established the asymptotic convergence of a QMC sequence in terms of the training set size, while Jank (2005) proposed replacing MC with QMC in computing the E-step of the EM algorithm. Similar to our motivation, Buchholz et al. (2018) analyzed the convergence of SGD when samples are drawn using QMC. Their approach differs significantly from ours in that their method draws an independent unbiased length-$b$ QMC sequence for each minibatch, while our examples come from contiguous subsequences of a length-$T$ QMC sequence that is used across *all* iterations.

## 3 EXAMPLE-GRADIENT AVERAGES AND SGD CONVERGENCE

In this section, we describe our setup, define the average gradient error condition we are proposing, and state our main result. Our objective is to minimize a continuously differentiable function $f : \mathbb{R}^d \to \mathbb{R}$ using examples $x_t$ from some set $\mathcal{X}$ retrieved at each iteration $t$. We make the usual assumption that both the loss gradient and the example gradients are $L$-Lipschitz continuous.

**Assumption 1** (*L-Smoothness*). *For some $L < \infty$, for any $u, v \in \mathbb{R}^d$ and any example $x \in \mathcal{X}$,*

$$\|\nabla f(u; x) - \nabla f(v; x)\| \leq L \cdot \|u - v\| \quad and \quad \|\nabla f(u) - \nabla f(v)\| \leq L \cdot \|u - v\|.$$

We propose a new condition on the *average gradient error*. Informally, this condition bounds how averages of consecutive example gradients approximate the objective gradient. Formally,

**Assumption 2.** *In the context of Equation (1), we say that the example sequence $x_0, x_1, \ldots$ is $(\gamma, C, \Phi)$-concentrating for $\gamma \in [1, 2]$, $C > 0$, $\Phi \geq 0$, if for any timestep $\tau \geq 0$ and any $m > 0$,*

$$\left\| \frac{1}{m} \sum_{t=\tau}^{\tau+m-1} \nabla f(w_\tau; x_t) - \nabla f(w_\tau) \right\|^2 \leq \frac{1}{m^\gamma} \left( C^2 + \Phi^2 \|\nabla f(w_\tau)\|^2 \right), \tag{2}$$

*where $w_\tau$ is the weight parameter vector arrived at after $\tau$ SGD update steps in Equation (1).*

The constants $C$ and $\Phi$ here may depend on the total number of iterations $T$: specifically, they may absorb logarithmic factors like $\log(\tau + m)^{2s}$ typical in a QMC error bound, where $s$ is the dimension of the sample space. In addition, when the $\{x_t\}$'s are random, we will show that this assumption holds with high probability. Furthermore, it suffices to show that the inequality holds for *any* $w$: our requirement that it holds only for the specific $w_\tau$ arrived at by SGD is weaker. We can develop intuition about Assumption 2 by considering familiar cases:

- For general with-replacement sampling, the sum in (2) is a sum of independent random variables; so, a concentration argument would yield Assumption 2 with $\gamma = 1$ with high probability.
- For general without-replacement sampling from a dataset of size $n$, any whole-epoch subsequences from the sum in (2) will cancel to zero, so we expect that sum to have magnitude $\mathcal{O}(n)$ independent of $m$, and thus Assumption 2 should hold with $\gamma = 2$ (Propositions 1 to 4).
- For low-discrepancy sampling, we would expect Assumption 2 to hold with $\gamma = 2$, along with a multiplicative $\log(\tau + m)^{2s}$ term: this is the classic error rate for QMC (Proposition 6).

In Section 4, we will make this intuition rigorous with high probability under the bounded gradient error assumption. This intuition illustrates both how Assumption 2 can cover previously analyzed settings and how it can generalize to cases not previously studied, such as QMC data augmentation. Also observe that Assumption 2 is easily adapted to minibatch SGD: if it holds for the single-example case, it should also hold with modified constants for minibatches of size $b$ (consisting of averages of $b$ consecutive example gradients) by substituting $m \mapsto mb$.[1] Since this is straightforward, for simplicity of presentation our theory focuses on the batch-size-1 case. Our analysis is based on the following key lemma, which bounds the evolution of SGD over an "analysis phase" of $m$ steps.

**Lemma 1.** *Suppose our setup satisfies Assumptions 1 and 2 and that we use a constant step size $\alpha$. For all timesteps $\tau \geq 0$, let $m > 0$ be some integer such that $3\alpha m^{1-\gamma/2}\Phi \leq \alpha m \leq \frac{1}{6L}$. Then the objective at timestep $\tau + m$ is bounded by*

$$f(w_{\tau+m}) \leq f(w_\tau) - \tfrac{1}{4}\alpha m \|\nabla f(w_\tau)\|^2 + 2\alpha m^{1-\gamma} C^2.$$

By applying this lemma inductively and choosing the step size appropriately, we can derive convergence rates for SGD in a variety of settings. Note that while here for simplicity we state results for a fixed step size, Lemma 1 can also be used to get essentially the same rates for diminishing step size schemes, which we analyze in Appendices A.2 and A.3. In what follows, we let $f^*$ be the global minimum of $f$, let $\Delta := f(w_0) - f^*$, and use $\tilde{\mathcal{O}}$ to hide logarithmic terms in the problem parameters such as $C$, $\Phi$, $L$, $\Delta$, and $\epsilon$, while treating $\gamma$ as a constant.

**Theorem 1** (*Non-convex case*). *Suppose that our setup satisfies Assumptions 1 and 2, and let $\epsilon > 0$ be any target error. Using SGD (1) with a constant step size $\alpha = \frac{1}{6L} \left\lceil (4C/\epsilon + 3\Phi)^{2/\gamma} \right\rceil^{-1}$, the number of steps $T$ needed to achieve $\min_{t=0,\cdots,T-1} \|\nabla f(w_t)\|^2 \leq \epsilon^2$ is at most*

$$T = \left\lceil \tfrac{48L\Delta}{\epsilon^2} \right\rceil \cdot \left[ \left( \tfrac{4C}{\epsilon} + 3\Phi \right)^{2/\gamma} \right] = \tilde{\mathcal{O}} \left( \tfrac{C^{2/\gamma} L\Delta}{\epsilon^{2+2/\gamma}} + \tfrac{\Phi^{2/\gamma} L\Delta}{\epsilon^2} + \tfrac{C^{2/\gamma}}{\epsilon^{2/\gamma}} + \Phi^{2/\gamma} \right).$$

---

[1] The only technical subtlety is that the iterates $w_\tau$ of SGD would vary with minibatch sizes.

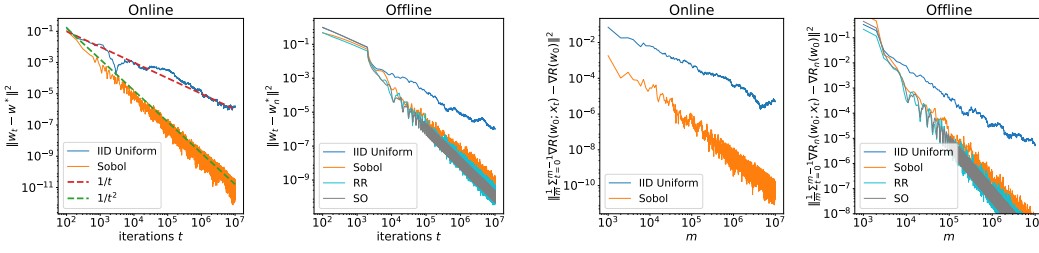

|  |  |
|---|---|
| (a) Distance to optimum over iterations | (b) Average gradient error (LHS of Assumption 2) |

Figure 1: Comparison of sampling schemes on a synthetic least squares problem.

Observe that when $\gamma = 1$, this gives the standard $\epsilon^{-4}$ rate expected for non-convex SGD, and for $\gamma = 2$, this gives us the accelerated $\epsilon^{-3}$ rate of shuffling methods. We can get an even faster rate if $f$ satisfies the $\mu$-Polyak-Łojasiewicz (PL) condition $\|\nabla f(w)\|^2 \geq 2\mu(f(w) - f^*)$, which generalizes strong convexity and has been applied before in the study of sample orders (Mishchenko et al., 2020; Ahn et al., 2020): in particular the following theorem holds for $\mu$-strongly convex functions.

**Theorem 2.** *Suppose that $f$ satisfies the $\mu$-PL condition and our setup satisfies Assumptions 1 and 2. Let $\epsilon > 0$ be any target error, and $\kappa = L/\mu$ be the condition number of the problem. Using SGD (1) with a constant step size $\alpha = \frac{1}{6L}\left\lceil (8C^2/(\mu\epsilon^2) + 9\Phi^2)^{1/\gamma} \right\rceil^{-1}$, the number of steps $T$ needed to guarantee $f(w_T) - f^* \leq \epsilon^2$ is at most*

$$T = \left\lceil 12\kappa \log\left(\frac{2\Delta}{\epsilon^2}\right) \right\rceil \cdot \left\lceil \left(\frac{8C^2}{\mu\epsilon^2} + 9\Phi^2\right)^{1/\gamma} \right\rceil = \tilde{\mathcal{O}}\left(\frac{C^{2/\gamma}\kappa}{\mu^{1/\gamma}\epsilon^{2/\gamma}} + \kappa\Phi^{2/\gamma} + \kappa\right).$$

Observe that when $\gamma = 1$, this recovers the ordinary $T = \kappa\epsilon^{-2}$ rate we usually get for strongly convex SGD, and when $\gamma = 2$ we get a faster rate of $\kappa\epsilon^{-1}$. These theorems together show that our Assumption 2 is sufficient to show the convergence of SGD, and the convergence-rate parameter $\gamma$ of the assumption translates to affect the convergence rate of SGD. This validates our intuition that *faster convergence of averages of consecutive example gradients to the full gradient $\nabla f(w)$ leads to faster convergence of SGD.*

**Synthetic experiments.** We quickly validate these results on a synthetic 10-dimensional strongly-convex problem. The first setting we consider is the *expected risk minimization* of $R(w) = \mathbb{E}\left[(x^\intercal w - y)^2\right]$, where $x \sim \mathcal{N}(0, I_d)$ and $y \,|\, x \sim \mathcal{N}(x^\intercal w^*, 1)$ for some optimal value $w^*$. We run SGD in an online fashion—at iteration $t$ we draw samples $(x_t, y_t)$ from the underlying distribution to compute $\nabla R(w_t; x_t)$ used in the update. We compare drawing these samples independently at random against drawing using a QMC sequence (Sobol in $[0, 1]^d$) via an inverse transform, using for both cases the same diminishing step size scheme selected to minimize the expected risk for the random-sampling case—the optimal step size scheme for vanilla SGD. The online plot of Fig. 1(a) shows that the convergence rate is strictly superior with QMC, which achieves a $\mathcal{O}(1/t^2)$ rate compared to the $\mathcal{O}(1/t)$ rate of random sampling, which is what Theorem 2 predicts.

We also evaluate the offline setting, where we draw $n$ independent examples from the same distribution to form a training set, and minimize the *empirical risk* $R_n(w) = \frac{1}{n}\sum_{i=1}^{n}(x_i^\intercal w - y_i)^2$. This corresponds to a least squares problem with optimal solution $w_n^*$. We run SGD epoch-wise for $K$ epochs, in which we compare sampling from the training set uniformly with replacement (IID Uniform), random reshuffling (RR), shuffle once (SO), and sampling using one QMC sequence in $[0, 1]$ of length $T = nK$ followed by a mapping to example indices (Sobol). In the offline plot of Fig. 1(a), we see that the low-discrepancy methods all yield an accelerated rate compared to IID Uniform, which again validates our theory. Additional details can be found in Appendix A.1.3.

With the same synthetic setup, we also verified our bound in Assumption 2 by measuring the average gradient error over a sequence of examples. The fixed point $w_\tau$ is arbitrarily set to the origin. For the online setting (left of Fig. 1(b)), we use the same set of examples as in the SGD experiments above starting at $t = 0$. Similarly, in the offline setting, we go through the examples epoch-wise. As we can see, the sample orderings given by QMC, RR and SO (offline only) indeed give us an accelerated rate of decrease in the average gradient error as we increase $m$ ($\gamma \approx 2$), justifying our main assumption.

## 4 ANALYSIS OF EXISTING SCAN ORDERS

In this section, we illustrate the power of our approach by proving convergence rates for example-selection methods proposed and analyzed in previous literature. For each method, we show Assumption 2 holds with high probability, and then by applying Theorems 1 and 2 to these results, we show how existing rates for these methods can be recovered and in some ways improved upon. To the best of our knowledge, these are the first high-probability results for shuffle once and random reshuffle for general non-convex optimization. When using a finite training set of examples, we let $n$ denote its size, let the examples be indexed as $x^{(0)}, x^{(1)}, \ldots, x^{(n-1)}$ (to avoid confusion with $x_t$, the example used by SGD at step $t$), and let $T = nK$ denote the total number of iterations after $K$ epochs. We will also require the following standard assumption that bounds the error of a single example gradient.

**Assumption 3.** *For all examples $x \in \mathcal{X}$ and points $w \in \mathbb{R}^d$, there exists $A, B \geq 0$ such that the gradient errors satisfy $\|\nabla f(w; x) - \nabla f(w)\|^2 \leq A^2 + B^2 \|\nabla f(w)\|^2$.*

**Shuffle once (SO).** In the shuffle-once variant of SGD, a single permutation $\sigma$ of $\{0, \ldots, n-1\}$ is chosen uniformly at random at the start, and the examples are used repeatedly in that order: explicitly, $x_t = x^{(\sigma(t \bmod n))}$. One way to analyze shuffle-once is to prove Assumption 2 for permutation-based methods generally.

**Proposition 1.** *Let Assumptions 1 and 3 hold, and suppose that we are using a permutation-based method of sampling, that is, any method such that $x_{kn}, x_{kn+1}, \ldots, x_{kn+n-1}$ is a permutation of $x^{(0)}, x^{(1)}, \ldots, x^{(n-1)}$ for all epochs $k \geq 0$. Any such method satisfies Assumption 2 with $\gamma = 2$, $C^2 = n^2 A^2$ and $\Phi^2 = n^2 B^2$.*

This immediately lets us recover previous rates up to constant factors, but we can do better. In the case where we are learning over a bounded region, we can prove a stronger result for shuffle once.

**Proposition 2.** *Suppose that we are using the shuffle once variant of SGD to learn over a region $\mathcal{B} \in \mathbb{R}^d$ of radius at most $R$, such that the iterates $w_t$ are guaranteed to remain within this region. Assume that for all $w \in \mathcal{B}$ and all examples $x$ in the training set of size $n$, Assumption 1 (L-Smoothness) and Assumption 3 hold. Then with probability at least $1 - p$, Assumption 2 holds with $\gamma = 2$, $C^2 = \tilde{\mathcal{O}}(dA^2(n + B^2))$, and $\Phi^2 = \tilde{\mathcal{O}}(ndB^2)$.*[2]

In comparison to previous results and to our rate implied by Proposition 1, this improves the dependence from $n$ to $\sqrt{nd}$, which is a significant improvement over the best rates for shuffle once available in the literature when the dimension is small relative the training set size. In particular, if we set $B = 0$ and consider small $\epsilon$, our rate in the non-convex case becomes

$$T = \tilde{\mathcal{O}}\left(\frac{AL\Delta\sqrt{nd}}{\epsilon^3}\right), \quad \text{which implies} \quad \epsilon^2 \leq \tilde{\mathcal{O}}\left(\frac{(AL\Delta)^{2/3}(nd)^{1/3}}{T^{2/3}}\right) = \tilde{\mathcal{O}}\left(\left(\frac{d}{nK^2}\right)^{1/3}\right).$$

This rate matches that for shuffle once achieved in Nguyen et al. (2020, Corollary 1) in terms of $\epsilon$, up to logarithmic factors. In the $\mu$-PL (or strongly convex) case for $B = 0$, we again obtain a rate matching that of Nguyen et al. (2020), Ahn et al. (2020), and Mishchenko et al. (2020):

$$T = \tilde{\mathcal{O}}\left(\frac{\kappa A}{\epsilon}\sqrt{\frac{nd}{\mu}}\right), \quad \text{which implies} \quad \epsilon^2 \leq \tilde{\mathcal{O}}\left(\frac{\kappa^2 A^2 nd}{\mu T^2}\right) = \tilde{\mathcal{O}}\left(\frac{\kappa^2 A^2 d}{\mu nK^2}\right).$$

**Random reshuffling (RR).** Random reshuffling is similar to shuffle once, except that a new ordering is chosen at each epoch: sampling the dataset without replacement. Concretely, if $\sigma_k$ denotes the permutation used by random reshuffling at the $k$th epoch, then $x_t = x^{(\sigma_{\lfloor t/n \rfloor}(t \bmod n))}$.

**Proposition 3.** *Suppose that we are using the random reshuffling variant of SGD. Assume that for all $w \in \mathbb{R}^d$ and all examples, Assumption 1 (L-Smoothness) and Assumption 3 hold. For some $p \in (0, 1)$, set the constant step size to satisfy $\alpha \leq \left(\max\left\{1460 BnL \cdot \log\left(4e^2 T/p\right), 2nL\right\}\right)^{-1}$. Then with probability at least $1 - p$, Assumption 2 holds with $\gamma = 2$, $C^2 = \tilde{O}(nA^2)$ and $\Phi^2 = \tilde{O}(nB^2)$.*

Setting $B = 0$, our rate in the non-convex case now becomes

$$T = \tilde{\mathcal{O}}\left(\frac{AL\Delta\sqrt{n}}{\epsilon^3}\right), \quad \text{which implies} \quad \epsilon^2 \leq \tilde{\mathcal{O}}\left(\frac{(AL\Delta)^{2/3}(n)^{1/3}}{T^{2/3}}\right) = \tilde{\mathcal{O}}\left(\left(\frac{1}{nK^2}\right)^{1/3}\right).$$

---

[2]The probability $p$ is only present in log terms in these expressions, so it does not appear in the $\tilde{\mathcal{O}}$.

Here we match the best rate obtained by Mishchenko et al. (2020, Corollary 3) in the small $\epsilon$ setting. In the $\mu$-PL (or strongly-convex) case, we get

$$T = \tilde{\mathcal{O}}\left(\frac{\kappa A}{\epsilon}\sqrt{\frac{n}{\mu}}\right), \quad \text{which implies} \quad \epsilon^2 \le \tilde{\mathcal{O}}\left(\frac{\kappa^2 A^2 n}{\mu T^2}\right) = \tilde{\mathcal{O}}\left(\frac{\kappa^2 A^2}{\mu n K^2}\right).$$

As in shuffle once, here our rate for random reshuffling matches that obtained by Nguyen et al. (2020) and Ahn et al. (2020), as well as Mishchenko et al. (2020) albeit with a slightly worse dependency on $\kappa$. It's worth noting that for simple quadratics, RR and SO are only faster than with-replacement SGD when $K \gtrsim 1/\mu$ (Safran & Shamir, 2021).

**Random reshuffling with data echoing.** Data echoing is a technique that can be easily implemented in a machine learning training pipeline to increase throughput and improve performance. It was first introduced and tested empirically by Choi et al. (2019) and analyzed by Agarwal et al. (2020). The idea is to perform multiple SGD updates on each example $x_i$ (or minibatch) before proceeding to the next. By "echoing" examples we allow more time for upstream data loading and preprocessing, and consequently decrease downstream GPU idle time for gradient computation. For simplicity, we also use a fixed number of echos $c$ as in Agarwal et al. (2020). Concretely, a $c$-echoed version of a sample order $\hat{x}$ is given by $x_t = \hat{x}_{\lfloor t/c \rfloor}$. Data echoing can essentially be applied to any example-ordering scheme, and here we provide one analysis under random reshuffling. The justification for Assumption 2 under random reshuffling with data echoing follows essentially without modification from the $c = 1$ version in the previous subsection.

**Proposition 4.** *Suppose that we are using the random reshuffling variant of SGD, where each example is echoed $c$ times using the same step size in RR. Under the same assumptions as in Proposition 3, with probability at least $1 - p$, Assumption 2 holds with $\gamma = 2$, $C^2 = \tilde{O}(cnA^2)$ and $\Phi^2 = \tilde{O}(cnB^2)$.*

It immediately follows that data echoing should get the same convergence rates we showed in Section 4 with $A^2$ and $B^2$ multiplied by $c$. Although Agarwal et al. (2020) also provided an analysis for data echoing, they require that the examples are sampled independently, rather than the random reshuffling setting that is more commonly-used: as a result, their analysis did not achieve the accelerated $\epsilon^{-3}$ rate that shuffled methods enjoy. Another advantage of our analysis is that the proof follows exactly from that of vanilla random reshuffling, from which the constant $c$ simply propagates.

**Markov chain gradient descent (MCGD).** To illustrate the versatility of Assumption 2, we show how it can be satisfied by a problem where the objective is not a finite sum and where $\gamma \ne 2$. Consider $f(w) = \mathbb{E}_{\xi \sim \Xi}[f(w; \xi)]$ with some underlying distribution $\Xi$. Running SGD then requires that at each iteration $t$, we draw $\nabla f(w_t; \xi_t)$ where $\xi_t \sim \Xi$; however, sampling from $\Xi$ can be intractable. The method of Markov Chain Gradient Descent addresses this problem by sampling the $\xi_t$ from the trajectory of a single Markov chain with stationary distribution $\Xi$ (Sun et al., 2018). The intuition is that although at early iterations the $\xi$'s have not converged to their true distribution, the iterates visited by SGD are also far from the optimum, thus larger approximation error in the early $\xi$'s is rather harmless. As we continue iterating, the Markov chain will mix as SGD converges. We show that the convergence of MCGD can be bounded in terms of the mixing time of that Markov chain.

**Proposition 5.** *Suppose that we use samples $x_t$ from a Markov chain with mixing time $t_{mix}$. Assume that for all $w \in \mathbb{R}$ and all examples $x_t$, Assumption 3 holds. Then with probability at least $1 - p$, Assumption 2 holds with $\gamma = 1$, $C^2 = \tilde{\mathcal{O}}(A^2 t_{mix}^2)$, and $\Phi^2 = \tilde{\mathcal{O}}(B^2 t_{mix}^2)$.*

It follows that for non-convex optimization in the $B = 0$ case, our convergence rate is given by

$$T = \tilde{\mathcal{O}}\left(\frac{A^2 L \Delta t_{mix}^2}{\epsilon^4}\right), \quad \text{which implies} \quad \epsilon^2 \le \tilde{\mathcal{O}}\left(\frac{A t_{mix}\sqrt{L\Delta}}{T^{1/2}}\right).$$

Sun et al. (2018, Theorem 2) use a diminishing step size $O(1/t^q)$ to obtain $T = O(\epsilon^{-2/1-q})$, where $q \in (1/2, 1)$. In contrast, our rate is faster and holds with high probability instead of in expectation.

## 5 NEW EXAMPLE-SELECTION METHODS FOR FASTER CONVERGENCE

The analysis in Section 4 focused on recovering the convergence rates for SGD with known example-ordering algorithms. In this section, we propose two new example-selection approaches that allow faster convergence: QMC-based data augmentation and greedily minimizing the metric in (2).

---

**Algorithm 1** Example-Ordered SGD via Greedily Minimizing Average Gradient Error

---

**Input:** step size $\alpha$, number of iterations $T$, random projection matrix $\Pi$, buffer for gradients estimation: $g_i \leftarrow 0, \forall i \in \{0, \cdots, n-1\}$, $g \leftarrow 0$, initial weights $w_0$, initial permutation $\sigma_0$.

1: **for** $t = 0, \cdots, T/n - 1$ **do**
2:      Initialize : $g_i \leftarrow 0, \forall i \in \{0, \cdots, n-1\}$; $g \leftarrow 0$; $\mathcal{I} \leftarrow \emptyset$.
3:      **for** $i = 0, \cdots, n-1$ **do**
4:          Update the model parameters: $w_{tn+i+1} \leftarrow w_{tn+i} - \alpha \nabla f(w_{tn+i}; x^{(\sigma_t(i))})$.
5:          Update the buffers: $g_{\sigma_t(i)} \leftarrow \Pi \nabla f(w_{tn+i}; x^{(\sigma_t(i))})$; $g \leftarrow g + g_{\sigma_t(i)}$.
6:      **end for**
7:      **for** $i = 0, \cdots, n-1$ **do**
8:          $\sigma_{t+1}(i) \leftarrow \underset{i \in \{0, \cdots, n-1\} \setminus \mathcal{I}}{\arg\min} \left[ \left\| \sum_{j \in \mathcal{I} \cup \{i\}} (g_j - g/n) \right\|^2 \right]$; $\mathcal{I} \leftarrow \mathcal{I} \cup \{\sigma_{t+1}(i)\}$.
9:      **end for**
10: **end for**
11: **return** $w_T$

---

**QMC-based data augmentation.** In many scenarios where only limited examples are given, we want to augment the dataset for better generalization. More formally, given a transform function $\mathcal{A}$ that takes example $x$ and a random variable $\zeta$ uniformly distributed in $[0, 1]^s$ as input, where $s$ denotes the dimension of augmentation space, the augmented objective for a dataset $\mathcal{D}$ of size $n$ is

$$f(w) = \frac{1}{n} \sum_{x \in \mathcal{D}} \mathbb{E}_{\zeta \sim \mathcal{U}[0,1]^s} f(w; \mathcal{A}(x, \zeta)) = \frac{1}{n} \sum_{x \in \mathcal{D}} \int_{\mathbb{R}^s} f(w; \mathcal{A}(x, \zeta)) \, d\zeta. \tag{3}$$

The rationale is that by performing some reasonable random transformation on a given example, we assume the output would be another example that is identically distributed, and the expected value models an infinitely-large training set consisting of such transformed examples. For example, in an image classification task, we could set $s = 1$ and have $\mathcal{A}(x, \zeta)$ output the image $x$ rotated by an angle of $20°(2\zeta - 1)$, modeling that a slight rotation of an image should preserve its label.

QMC can approximate this expectation with a low-discrepancy sequence of $\zeta_t$ drawn from the $s$-dimensional unit cube $[0, 1]^s$. Examples of such sequences include the Halton and Sobol sequences (Drmota & Tichy, 2006). QMC is especially favorable for data augmentation because $s$ is usually small in most data augmentation methods. We propose to use QMC for data augmentation together with random reshuffling. Concretely, if $\zeta_0, \zeta_1, \ldots$ is our low-discrepancy sequence and $\sigma_k$ denotes the permutation used by random reshuffling in the $k$th epoch, then we propose to use the example $x_t = \mathcal{A}(x^{(\sigma_{\lfloor t/n \rfloor}(t \bmod n))}, \zeta_{\lfloor t/n \rfloor + \sigma_{\lfloor t/n \rfloor}(t \bmod n)})$. That is, when we sample example $i$ in epoch $k$, we use the $(k + i)$th element of the low-discrepancy sequence. This is not the only reasonable way of combining QMC and RR: it is just one way we found to work well. In theory we would expect the approximation error here to decay at the rate $\tilde{\mathcal{O}}(1/m^2)$, instead of the $\mathcal{O}(1/m)$ of the random sampling that is standard for data augmentation. To prove this rigorously, we make two additional assumptions which are commonly used in analyzing QMC sequences (Aistleitner & Dick, 2014).

**Assumption 4** (Bounded gradient variation). *There exists a constant $V > 0$ such that for any fixed $w \in \mathbb{R}^d$ and $x \in \mathcal{X}$, the example gradients as a function of $\zeta$ under QMC data augmentation, $F(\zeta) = \nabla f(w; \mathcal{A}(x, \zeta))$, has Hardy-Krause variation (Tezuka, 2000) at most $V$, that is $V_{HK}(F) \leq V$.*

**Assumption 5.** *The QMC sequence $\{\zeta_t\}_{t \geq 0}$ has low star-discrepancy (Owen, 2003): for all $m > 0$,*

$$\sup_{a \in [0,1]^s} \left| \frac{1}{m} \sum_{t=0}^{m-1} \mathbb{1}\{\zeta_t \in [0, a)\} - \prod_{j=1}^{s} a_j \right| \leq \mathcal{C}_{QMC} \cdot \frac{\log(m)^s}{m},$$

*where $[0, a) = \{x \in [0, 1]^s \mid 0 \leq x_j < a_j, j = 1, \ldots, d\}$ for some constant $\mathcal{C}_{QMC}$.*

**Proposition 6.** *Suppose that we are using the random reshuffling variant of SGD with QMC data augmentation as described. Assume that for all $w \in \mathbb{R}^d$ and all examples, Assumptions 1 , 3, 4 and 5 hold for Equation 3. For some $p \in (0, 1)$, set the step size to be a constant such that $\alpha \leq \left( \max\{1460 BnL \cdot \log(4e^2 T/p), 2nL\} \right)^{-1}$. Then with probability at least $1 - p$, Assumption 2 holds with $\gamma = 2$, $C^2 = \tilde{\mathcal{O}}(n^2 V^2 \mathcal{C}_{QMC}^2 \log(T)^{2s} + nA^2)$ and $\Phi^2 = \tilde{\mathcal{O}}(nB^2)$.*

Comparing it with Proposition 3, this QMC variant enjoys the same $\tilde{\mathcal{O}}(1/T^2)$ rate we get for vanilla random reshuffling: to our knowledge, this is the first accelerated rate for learning with data augmen-

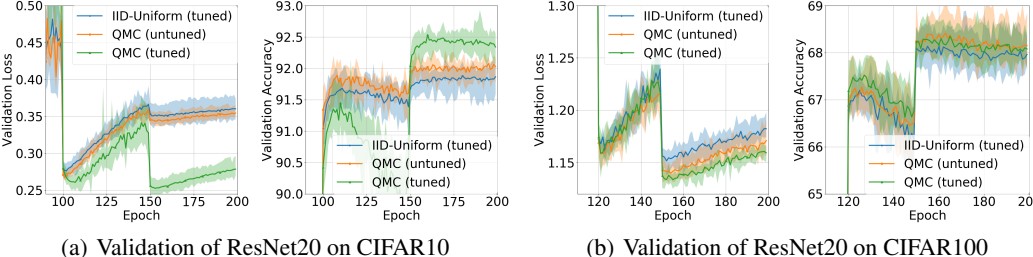

(a)  Validation of ResNet20 on CIFAR10

(b)  Validation of ResNet20 on CIFAR100

Figure 2: Data augmentation with IID-uniform (standard) and QMC-based methods on CIFAR.

tation. In addition to the theoretical advance, we demonstrate in Section 6 that our QMC variant can achieve better validation performance in practice in multiple applications.

**Better example ordering via greedy selection.** Taking a closer look at Assumption 2, the magnitude of the left-hand side plays a crucial role in the convergence. In the ERM setting, this motivates us to select a permuted example order that minimizes this expression, following which SGD would converge with minimized average gradient error. However, naively constructing such a sequence is tedious as iterating over all the $\tau, m > 0$ and all permutations can be computationally intensive. In light of this, we apply several approximation techniques into the construction and formulate it into Algorithm 1. The first technique is to use *stale gradients*, i.e., using the gradients computed at each epoch to estimate the sequence used in the next epoch (line 5 in Algorithm 1). The intuition is that based on the smoothness of loss function (Assumption 1), we would expect the stale gradients to approximate the current gradients with tolerant approximation error as long as the step size is reasonably small. The second technique is *random projection* (line 9): we search a lower-dimensional space, which allows faster construction and reduces memory use (see similar strategies of using smoothness and dimension reduction techniques in (Caflisch, 1998)). Because this is a selection-by-permutation method, under Assumption 3 our greedy selection method will trivially satisfy Assumption 2 with $\gamma = 2$, $C = nA$, and $\Phi = nB$ (see Proposition 1).

## 6 EXPERIMENTS

In this section we evaluate our new algorithms on several deep learning benchmarks. First, we compare QMC-based data augmentation against IID-uniform augmentation on CIFAR10/100 and ImageNet datasets. Second, we compare greedy ordering (Algorithm 1) with RR and SO, and show how to further accelerate it with randomly projected sorting. Other details on the experimental setup can be found in Appendix A.1.1. Each experiment is repeated 10 times with consistent seeds among the algorithms.

Table 1: Top1 validation accuracy (%) of ResNet18 on ImageNet. The original one is the standard benchmark provided by PyTorch.

| Original | Uniform (tuned) | QMC (untuned) |
|----------|-----------------|---------------|
| 69.76    | 70.19           | **70.48**     |

**QMC-based data augmentation.** We start by training ResNet20 on CIFAR10 and CIFAR100, where discrete and continuous data augmentations are applied, respectively. Specifically, CIFAR10 uses random crop and random horizontal flip while CIFAR100 uses an additional random rotation of 15 degrees. To apply the QMC-based data augmentation, we first generate a Sobol sequence of appropriate dimension using the `qmcpy` package (Choi et al., 2020+), and then replace the pseudorandom sequence used in the original random augmentation pipelines with that Sobol sequence. We run the baseline IID-uniform method with finetuned hyperparameters (weight decay $10^{-4}$), which reproduces the result from He et al. (2016) with an error rate 8.4%. Then we run QMC-base augmentation with the same hyperparameter (untuned) and finetuned counterparts, with a grid search over weight decay values in $\{r \cdot 10^{-4}\}_{r=1}^{4}$. From Figure 2 we observe the QMC-based augmentation consistently outperforms the baseline methods, even without hyperparameter tuning. Comparing Figure 2 with He et al. (2016), we observe the QMC-based augmentation allows ResNet20 to reach comparable validation accuracy as ResNet44 while requiring only 40% as many parameters (0.27M vs 0.66M). We run a $t$-test on these results (in Appendix A.1.2) to show the validation accuracy from the two augmentation methods are different statistically significantly (p-value $p = 7 \cdot 10^{-7}$ on

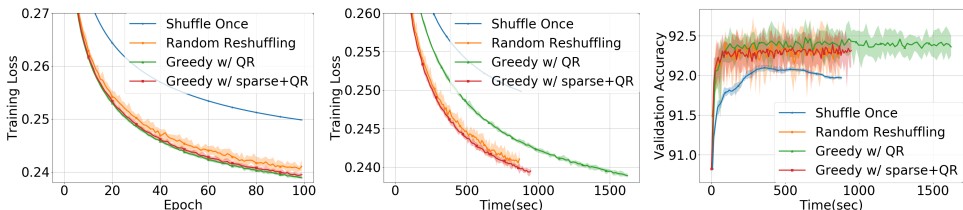

(a) Logistic Regression on MNIST. The greedy algorithm reuses the hyperparameters finetuned on RR.

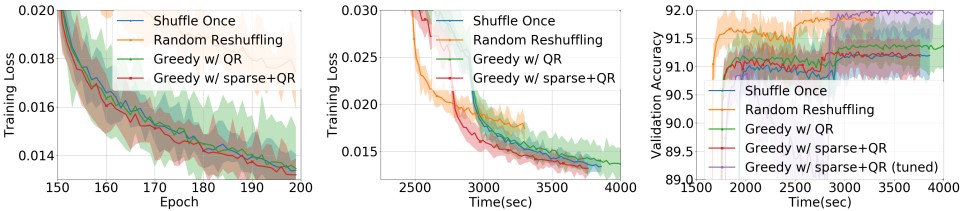

(b) ResNet20 on CIFAR10. The greedy algorithm is able to converge faster when optimizing the same loss function while achieving SOTA validation accuracy with mild tuning.

Figure 3: Comparison between Algorithm 1 and RR/SO on MNIST and CIFAR10.

CIFAR10 and $p$=0.036 on CIFAR100). Importantly, this improvement comes essentially for free, as generating low discrepancy sequences in low dimension has very little overhead.

We also evaluate our method on fine-tuning ResNet18 on ImageNet. We apply different augmentation methods on a pre-trained model and train for 5 additional epochs with step size $10^{-4}$. We report their Top1 accuracies in Table 6, which shows that our QMC method improves the validation accuracy by 0.3% compared to the same number of epochs of fine-tuning using random sampling.

**Better example ordering via greedy selection.** In this section we evaluate Algorithm 1 on two benchmarks: Logistic Regression on MNIST and ResNet20 on CIFAR10. As discussed, sorting the stale gradients naively could incur substantial overhead on memory and computation. To mitigate this, we adopt two methods: random projection and QR decomposition. The former is mainly to reduce storage: we obtain a gradient computed at some time in an epoch and project it into a lower-dimensional space before storing it. Classic ways of projection include Gaussian projection or random sparsification: we adopt the latter as it does not require storing the projection matrix, which minimizes the storage cost. After we obtain all the stale gradients, we concatenate them into a matrix and perform QR decomposition before sorting, which allows us to sort in a low-dimensional space while preserving the order of gradients since the inner products between any two tensors will remain the same Gander (1980). We set the target dimension to be of 10% size of the original space. In the spirit of evaluating the applicability of Algorithm 1, we do not perform hyperparameter tuning in this section but reuse the ones tuned in the literature on Random Reshuffling.

We plot the results in Figure 3. In Figure 3(a) we observe greedy algorithms can consistently converge faster than RR and SO epoch-wise with optimizing the same loss function. When QR is used without projection, the algorithm is able to reach higher validation accuracy but converges slower with respect to the wall-clock time. On the other hand, when we use random sparsification additionally, the algorithm converges faster with respect to both epoch and wall-clock time without compromising the validation accuracy. For CIFAR10, we observe the greedy method can converge faster when optimizing the same loss as other baselines, and achieves higher validation accuracy when fine-tuned.

# 7 CONCLUSION

We present a unified analysis on example orderings used in SGD, which generalizes several widely-used orderings in the literature. We propose a greedy algorithm that allows faster convergence via constructing a better example order with approximate sorting techniques, as well as QMC-based augmentation that achieves higher validation accuracy on multiple benchmarks. One potential future direction is designing example orderings to more efficiently minimize the average gradient errors.

ACKNOWLEDGMENTS

The authors would like to thank A. Feder Cooper and anonymous reviewers from ICLR 2022 for their valuable feedbacks on earlier versions of this paper.

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

# A APPENDIX

## TABLE OF CONTENTS

## A.1 EXPERIMENT DETAILS

### A.1.1 ADDITIONAL DETAILS FOR SECTION 6

In Section 6, all the training scripts are implemented via PyTorch1.6 and run on a single machine configured with an 2.6GHz 4-core Intel (R) Xeon(R) CPU, 16GB memory and NVIDIA GeForce GTX 1080Ti with CUDA 10.1.

In the example ordering comparison, we use the same seed among different algorithms in the same run so as to guarantee every algorithm works with the same loss function.

In the ImageNet experiment, the standard step size schedule for ImageNet training is starting at 0.1 and decaying by 10 every 30 epochs. The pre-trained model is the trained model at epoch 90. Naturally, our learning rate should be 1e-4 by the same schedule. Other hyperparameters are adopted by the open source implementation: https://github.com/pytorch/examples/tree/master/imagenet.

In the data augmentation comparison, obviously other augmentation techniques can be used. The strategies we used are taken from open source implementation https://github.com/akamaster/pytorch_resnet_cifar10 and https://github.com/weiaicunzai/pytorch-cifar100 that can reproduce the validation accuracy in He et al. (2016), so that our comparison can be consistent with the correct benchmarks. Below we include the convergence plot for the experiment on ImageNet with ResNet18, where each algorithm is repeated three times with seeds uniformly selected from [0, 1000].

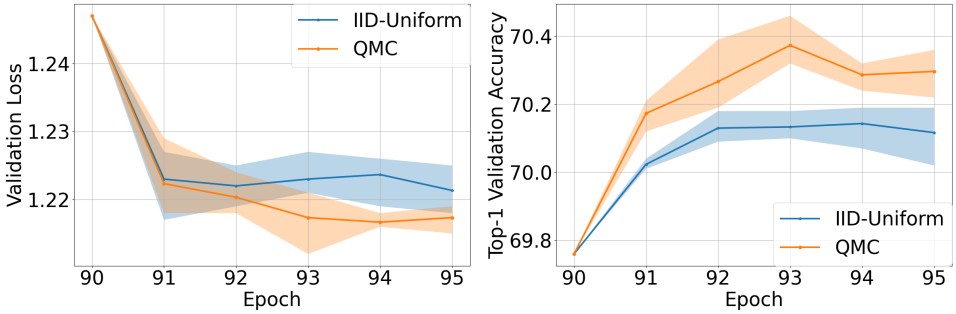

Figure 4: Data augmentation with IID-uniform (standard) and QMC-based methods on ImageNet.

### A.1.2 $t$-TEST FOR DATA AUGMENTATION RESULTS

We now perform t-test on the validation accuracy on two data augmentation results, to show the improvement from QMC is statistically significant. We include validation accuracy at epoch 200 for CIFAR10 and best accuracy for CIFAR100 in Table A.1.2. We compute the p-value between IID-Uniform and tuned QMC, and found the p-value on CIFAR10 and CIFAR100 to be 7e-7 and 0.036, respectively. Since they are both smaller than 0.05, we reject the null hypothesis which concludes they are of the same mean.

### A.1.3 SMALL EXPERIMENTS

For both experiments in Section 3, we generated $w^*$ from a standard Normal distribution. The minibatch size is 1, and the total number of iterations is $10^7$, with $n = 10^4$ over 1000 epochs for the offline setting. For all variants, we use the theoretical step size optimized for SGD with replacement (corresponding to IID Uniform) on this particular problem, given by

$$\alpha_{t+1} = \frac{\alpha_t(1 - \alpha_t)}{1 - \alpha_t^2(d + 2)} \quad \text{with} \quad \alpha_0 = \frac{\|w^*\|^2}{\|w^*\|^2(d + 2) + d},$$

which are derived in Appendix A.1.4 below. To obtain low-discrepancy samples from the Gaussian distribution, we use the inverse transform method. First we obtain the QMC sequences (Sobol in our

Table 2: Validation accuracy for different data augmentation methods on CIFAR datasets.

| | CIFAR10 | | | CIFAR100 | | |
| Runs | IID-Uniform | QMC | QMC (tuned) | IID-Uniform | QMC | QMC (tuned) |
|---|---|---|---|---|---|---|
| 1 | 91.87 | 92.05 | 92.45 | 67.92 | 68.91 | 68.21 |
| 2 | 91.67 | 92.13 | 92.53 | 68.01 | 68.92 | 68.12 |
| 3 | 91.68 | 92.03 | 92.33 | 68.03 | 68.53 | 68.42 |
| 4 | 91.88 | 91.9 | 92.35 | 67.82 | 68.03 | 68.71 |
| 5 | 92.05 | 92.03 | 92.41 | 68.89 | 68.41 | 67.99 |
| 6 | 91.79 | 91.93 | 92.15 | 66.99 | 68.66 | 68.51 |
| 7 | 91.91 | 92.79 | 92.59 | 68.31 | 68.92 | 68.53 |
| 8 | 91.53 | 91.73 | 92.44 | 68.14 | 68.09 | 68.53 |
| 9 | 91.3 | 91.96 | 92.21 | 68.02 | 67.91 | 68.12 |
| 10 | 91.82 | 92.21 | 92.16 | 67.73 | 68.31 | 68.53 |

case) $\zeta_t \in [0,1]^s$ where $s$ is the appropriate dimension, and then use that in the inverse CDF function of a Gaussian distribution to obtain the corresponding Gaussian sample.

In addition to using synthetic data, we also performed an offline version of the experiment Figure 1(b) on a real dataset, a6a from the LIBSVM repository (Chang & Lin, 2011). The dataset contains $n$=11220 examples with $d$=124 features (including bias), and all labels are binary. We use logistic regression with $\ell_2$ regularization (with $\lambda$=1e-4) as the empirical risk:

$$R_n(w) = \frac{1}{n} \sum_{i=1}^{n} \log(1 + \exp(-y_i x_i^\top w)) + \frac{\lambda}{2} \|w\|^2.$$

To measure the left hand side of Assumption 2, we again fix $w_\tau$ to be at the origin, and we take the number of epochs to be 1000. The results are in Figure 5 for one run with an arbitrarily-set seed, and we draw similar conclusions as observed with synthetic data (see Figure 1(b) in the main paper).

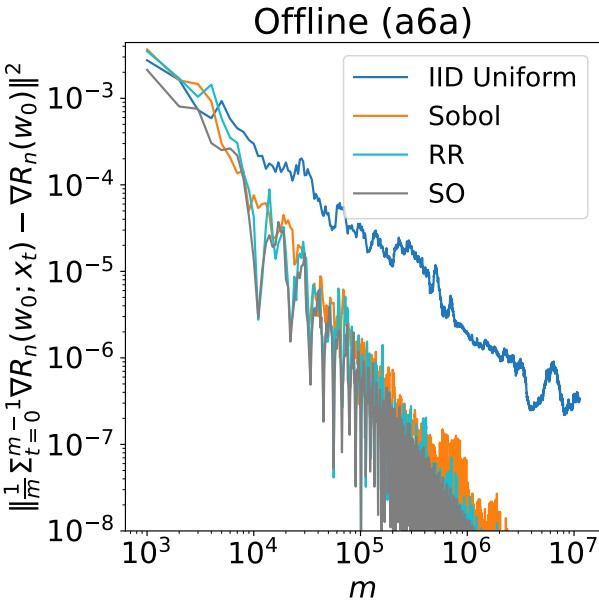

Figure 5: Comparison of sampling schemes on $\ell_2$-regularized logistic regression with real data.

### A.1.4 Optimal Step Size Derivation

In this section, we derive the optimal step-size sequence used in our synthetic toy example from Section 3. Recall our setup

$$x \sim \mathcal{N}(0, I_d), \quad y = x^\mathsf{T} w^* + \epsilon$$

where $w^*$ is the true model parameters that we are trying to recover, and $\epsilon \sim \mathcal{N}(0, 1)$ is some intrinsic error. The goal is to minimize the expected risk

$$f(w) = \frac{1}{2} \mathbb{E}\big[(x^\mathsf{T} w - y)^2\big].$$

By using the tower rule of expectation conditioning on $x$, this objective can be written as

$$f(w) = \frac{1}{2}(\|w - w^*\|^2 + 1).$$

The full gradient and the example gradient given a pair $(x, y)$ drawn from the above distribution are

$$\nabla f(w) = w - w^*,$$
$$\nabla f(w; x, y) = (x^\mathsf{T} w - y)x = (x^\mathsf{T}(w - w^*) - \epsilon)x.$$

Define the gradient error to be

$$\Delta^g := \nabla f(w; x, y) - \nabla f(w) = (xx^\mathsf{T} - I_d)(w - w^*) - \epsilon x. \tag{4}$$

Let us first analyze the expected gradient error conditioned on any randomness in $w$ that might arise from the sampling history:

$$\mathbb{E}\Big[\|\Delta^g\|^2\Big] = (w - w^*)^\mathsf{T} \mathbb{E}\big[(xx^\mathsf{T} - I_d)^2\big] - 2\mathbb{E}[\epsilon x^\mathsf{T}(xx^\mathsf{T} - I_d)](w - w^*) + \mathbb{E}[\epsilon^2 x^\mathsf{T} x].$$

The last term is simply $\mathbb{E}[\epsilon^2 x^\mathsf{T} x] = d$ since $\epsilon$ is independent of $x$. The middle term cancels to $0$ again by independence and zero-mean of $\epsilon$. For the first term,

$$\mathbb{E}\big[(xx^\mathsf{T} - I_d)^2\big] = \mathbb{E}[xx^\mathsf{T} xx^\mathsf{T} - 2xx^\mathsf{T} + I]$$
$$= (d + 2)I_d - 2I_d + I_d = (d + 1)I_d.$$

Together, we have

$$\mathbb{E}\Big[\|\Delta^g\|^2\Big] = (d + 1)\|w - w^*\|^2 + d. \tag{5}$$

Now let us derive an suboptimality gap recursion for the SGD update step using the step size $\alpha_t$:

$$\begin{aligned}
w_{t+1} - w^* &= w_t - \alpha_t \nabla f(w_t; x, y) - w^* \\
&= (w_t - w^*) + \alpha_t \nabla f(w_t) - \alpha_t \nabla f(w_t) - \alpha_t \nabla f(w_t; x, y) \\
&= (1 - \alpha_t)(w_t - w^*) - \alpha_t \Delta_t^g,
\end{aligned}$$

where $\Delta_t^g$ is the expression in Eq. (4) with $w_t$ as input. The expected suboptimality gap conditioned on all randomness up to timestep $t$ is then

$$\mathbb{E}\Big[\|w_{t+1} - w^*\|^2\Big] = (1 - \alpha_t)^2 \mathbb{E}\Big[\|w_t - w^*\|^2\Big] - 2(1 - \alpha_t)\mathbb{E}[((w_t - w^*)^\mathsf{T} \Delta_t^g] + \alpha_t^2 \mathbb{E}\Big[\|\Delta_t^g\|^2\Big].$$

Observe that under iid uniform sampling, $\mathbb{E}[\Delta_t^g] = 0$, and using Eq. (5) gives us

$$\mathbb{E}\Big[\|w_{t+1} - w^*\|^2\Big] = (1 - \alpha_t)^2 \|w_t - w^*\|^2 + \alpha_t^2(d + 1)\|w_t - w^*\|^2 + \alpha_t^2 d.$$

Taking an expectation over the entire history to remove conditioning and let $\rho_t := \mathbb{E}\Big[\|w_t - w^*\|^2\Big]$,

$$\begin{aligned}
\rho_{t+1} &= (1 - \alpha_t)^2 \rho_t + \alpha_t^2(d + 1)\rho_t + \alpha_t^2 d \\
&= (1 - 2\alpha_t + \alpha_t^2(d + 2))\rho_t + \alpha_t^2 d. \tag{6}
\end{aligned}$$

Note that the RHS is convex in $\alpha_t$, we can differentiate to minimize the expected suboptimality at every iteration $\rho_{t+1} = \mathbb{E}\Big[\|w_{t+1} - w^*\|^2\Big]$:

$$0 = (-2 + 2\alpha_t(d + 2))\rho_t + 2\alpha_t d \Rightarrow \alpha_t = \frac{\rho_t}{(d + 2)\rho_t + d}.$$

This also gives us an expression for $\rho_t$ in terms of $\alpha_t$:

$$\rho_t = \frac{d}{\alpha_t^{-1} - (d+2)},$$

which combined with Eq. (6) gives us

$$\rho_{t+1} = (1 - \alpha_t)\rho_t = (1 - \alpha_t)\frac{d}{\alpha_t^{-1} - (d+2)}.$$

Finally, the optimal step-size sequence can be implemented via the following recursion

$$\frac{1}{\alpha_{t+1}} = d + 2 + \frac{d}{\rho_{t+1}} = d + 2 + \frac{\alpha_t^{-1} - (d+2)}{1 - \alpha_t}$$

$$\frac{1}{\alpha_0} = d + 2 + \frac{d}{\rho_0} = d + 2 + \frac{d}{\|w_0 - w^*\|^2}.$$

It is easy to verify that this is indeed a decreasing sequence. If we initialize at $w_0 = 0$, then our step sizes are given by

$$\alpha_{t+1} = \frac{\alpha_t(1 - \alpha_t)}{1 - \alpha_t^2(d+2)} \quad \text{with} \quad \alpha_0 = \frac{\|w^*\|^2}{\|w^*\|^2(d+2) + d}.$$

## A.2 CONVERGENCE ANALYSIS: DIMINISHING STEP SIZE

In the main text, our convergence results focused on the constant step size regime for simplicity of presentation and ease of comparison to prior works, as they often lack diminishing step size results for permutation-based SGD variants. Here we formally present the convergence rate under our main Assumption 2 using a diminishing step size sequence. In Theorems 3 and 4 that we present below, the rate is of the same order as what we obtained under a constant step size (Theorems 1 and 2) up to a factor of $\log(T)$ that typically arises when using a diminishing step size.

Our analysis for the diminishing step size setting is accomplished by breaking the total number of iterations into phases. Suppose we want to run a total of $T$ iterations of SGD updates. We will break the analysis into $I$ number of phases. In each phase $i = 0, \ldots I - 1$, we run $K^{(i)} m^{(i)}$ number of iterations with constant step size $\alpha^{(i)}$. The step size is decayed at the beginning of each phase. For this, we define the triplets $(\alpha^{(i)}, m^{(i)}, K^{(i)})$, where $\{m^{(i)}\}$ is an increasing sequence.

Recall that $\gamma$ and $\Phi$ are constants in Assumption 2. In the *non-convex* case, the inner interval length is chosen to be

$$m^{(i)} = \lceil 3(\Phi + 1)^{2/\gamma} \rceil 2^i,$$

where as for the *strongly-convex* setting it is

$$m^{(i)} = \lceil 3(\Phi + 1)^{2/\gamma} \cdot e^{i/\gamma} \rceil.$$

For both function classes the diminishing step-size is set to

$$\alpha^{(i)} = \frac{1}{6Lm^{(i)}}. \tag{7}$$

### A.2.1 DIMINISHING STEP SIZE: NON-CONVEX CASE

**Theorem 3** (Diminishing step size: non-convex case). *Suppose that our setup satisfies Assumptions 1 and 2, and let $\epsilon > 0$ be any target error. After $T$ iterations of SGD (Eq. (1)) with a diminishing step-size (Eq. (7)), we can obtain*

$$\min_{t=0,\ldots,T-1} \|\nabla f(w_t)\|^2 \le 12 \log_2\left(\frac{16}{3(\Phi + 1)} T\right) \frac{L\Delta + C^2(3(\Phi + 1))^{-2}}{T^{\gamma/1+\gamma}(12(\Phi + 1)^2)^{-\gamma/1+\gamma} - 1} = \mathcal{O}\left(\frac{\log(T)}{T^{\gamma/1+\gamma}}\right).$$

*This implies the number of gradient evaluations to achieve $\min_t \|\nabla f(w_t)\|^2 \le \epsilon^2$ is*

$$T = \tilde{\mathcal{O}}(\epsilon^{\frac{-2(1+\gamma)}{\gamma}}).$$

*Proof.* To apply Lemma 1 to the iterations within a particular phase $i$, we need

$$3\Phi \le m^{\gamma/2} \qquad \text{and} \qquad \alpha \le \frac{1}{6Lm}$$

where we have temporarily dropped the index $(i)$ for convenience. Lemma 1 gives us the following bound on the objective value between any length-$m$ number of iterations within that phase: for $\tau \in \{0, m-1, 2m-1, \ldots, (K-1)m-1\}$,

$$f(w_{\tau+m}) \le f(w_\tau) - \frac{\eta}{4}\|\nabla f(w_\tau)\|^2 + \frac{2C^2}{\eta}\alpha^2 m^{2-\gamma}$$

$$\le f(w_\tau) - \frac{\alpha m}{4}\|\nabla f(w_\tau)\|^2 + 2C^2 \alpha m^{1-\gamma},$$

since the step size is constant throughout one phase. Summing over $K$ such intervals from $\tau = 0$ followed by a telescope,

$$\frac{\alpha m}{4} \sum_{k=0}^{K-1} \|\nabla f(w_{mk})\|^2 \le f(w_0) - f(w_{mK-1}) + 2C^2 K \alpha m^{1-\gamma}.$$

We now restore our phase index,

$$\frac{\alpha^{(i)}m^{(i)}K^{(i)}}{4}\frac{1}{K^{(i)}}\sum_{k=0}^{K^{(i)}-1}\|\nabla f(w_{mk})\|^2 \leq f(w_0) - f(w_{m^{(i)}K^{(i)}-1}) + 2C^2 K^{(i)}\alpha^{(i)}(m^{(i)})^{1-\gamma}$$

$$= f(w_0) - f(w_{m^{(i)}K^{(i)}-1}) + \frac{C^2}{3L}K^{(i)}(m^{(i)})^{-\gamma},$$

where we have chosen the largest possible $\alpha^{(i)} = 1/6Lm^{(i)}$ for all $i$. Summing over $I$ phases and letting $\Delta := f(w_0) - f^*$,

$$\sum_{i=0}^{I-1}\frac{\alpha^{(i)}m^{(i)}K^{(i)}}{4}\frac{1}{K^{(i)}}\sum_{k=0}^{K^{(i)}-1}\|\nabla f(w_{m^{(i)}k})\|^2 \leq I\Delta + \frac{C^2}{3L}\sum_{i=0}^{I-1}K^{(i)}(m^{(i)})^{-\gamma}$$

where the left hand side can be further lower bounded with

$$\sum_{i=0}^{I-1}\frac{\alpha^{(i)}m^{(i)}K^{(i)}}{4}\min_{t=0,\ldots,T-1}\|\nabla f(w_t)\|^2 \leq I\Delta + \frac{C^2}{3L}\sum_{i=0}^{I-1}K^{(i)}(m^{(i)})^{-\gamma},$$

using the inclusion that $\left\{0, m^{(i)}-1, \ldots, K^{(i)}m^{(i)}-1\right\}_{i=0}^{I-1} \subset \{0, \ldots, T-1\}$. Now let us choose

$$m^{(i)} = \lceil(3(\Phi+1))^{2/\gamma}\rceil 2^i$$
$$K^{(i)} = \lceil 2^{i\gamma}\rceil.$$

Here the choice of $m^{(i)}$ guarantees that

$$(m^{(i)})^{\gamma/2} = \left(\lceil(3(\Phi+1))^{2/\gamma}\rceil 2^i\right)^{\gamma/2}$$
$$= \lceil(3(\Phi+1))^{2/\gamma}\rceil^{\gamma/2}(2^i)^{\gamma/2}$$
$$\geq (3(\Phi+1))2^{i\gamma/2} \geq 3\Phi.$$

Next, observe that for the last term in the previous sum,

$$\sum_{i=0}^{I-1}K^{(i)}(m^{(i)})^{-\gamma} = \sum_{i=0}^{I-1}\lceil 2^{i\gamma}\rceil(\lceil(3(\Phi+1))^{2/\gamma}\rceil 2^i)^{-\gamma}$$
$$\leq \sum_{i=0}^{I-1}(2^{i\gamma}+1)((3(\Phi+1))^{2/\gamma}2^i)^{-\gamma}$$
$$\leq \sum_{i=0}^{I-1}2(2^{i\gamma})((3(\Phi+1))^{2/\gamma}2^i)^{-\gamma}$$
$$= \sum_{i=0}^{I-1}2(3(\Phi+1))^{-2}2^{i\gamma}2^{-i\gamma}$$
$$= 2I(3(\Phi+1))^{-2}.$$

Combining with the above,

$$\sum_{i=0}^{I-1}\frac{\alpha^{(i)}m^{(i)}K^{(i)}}{4}\min_{t=0,\ldots,T-1}\|\nabla f(w_t)\|^2 \leq I\Delta + \frac{2C^2}{3L}\cdot I(3(\Phi+1))^{-2}. \tag{8}$$

Furthermore, using our choice of $\alpha^{(i)}$

$$\sum_{i=0}^{I-1}\frac{\alpha^{(i)}m^{(i)}K^{(i)}}{4} = \frac{1}{6L}\sum_{i=0}^{I-1}K^{(i)}.$$

Re-arranging Eq. (8),

$$\min_{t=0,\dots,T-1} \|\nabla f(w_t)\|^2 \le \frac{6IL\Delta + \frac{2}{3}C^2 \cdot I(3(\Phi+1))^{-2}}{\sum_{i=0}^{I-1} K^{(i)}}. \tag{9}$$

The total number of iterations is given by

$$
\begin{aligned}
T &= \sum_{i=0}^{I-1} K^{(i)} m^{(i)} \\
&= \sum_{i=0}^{I-1} \lceil 2^{i\gamma} \rceil \cdot \lceil (3(\Phi+1))^{2/\gamma} \rceil 2^i \\
&\ge (3(\Phi+1))^{2/\gamma} \sum_{i=0}^{I-1} 2^{(\gamma+1)i} \\
&= (3(\Phi+1))^{2/\gamma} \frac{2^{(\gamma+1)I} - 1}{2^{\gamma+1} - 1} \ge (3(\Phi+1))^{2/\gamma} \frac{2^{(\gamma+1)I} - 1}{2^{\gamma+1}}.
\end{aligned}
$$

Solving for $I$ and using $\lg$ to denote $\log_2$, we obtain

$$I \le \left(\frac{1}{\gamma+1}\right) \lg \left(\frac{2^{\gamma+1}}{(3(\Phi+1))^{2/\gamma}} T + 1\right) \le \left(\frac{1}{\gamma+1}\right) \lg \left(\frac{2^{\gamma+2}}{(3(\Phi+1))^{2/\gamma}} T\right).$$

Moreover, we can also upper bound $T$ using similar arguments,

$$
\begin{aligned}
T &= \sum_{i=0}^{I-1} \lceil 2^{i\gamma} \rceil \cdot \lceil (3(\Phi+1))^{2/\gamma} \rceil 2^i \\
&\le \sum_{i=0}^{I-1} 2(2^{i\gamma}) \cdot 2(3(\Phi+1))^{2/\gamma} 2^i \\
&= 4(3(\Phi+1))^{2/\gamma} \sum_{i=0}^{I-1} 2^{(\gamma+1)i} \\
&= 4(3(\Phi+1))^{2/\gamma} \frac{2^{(\gamma+1)I}}{2^{\gamma+1} - 1},
\end{aligned}
$$

which gives

$$I \ge \left(\frac{1}{\gamma+1}\right) \lg \left(\underbrace{\frac{2^{\gamma+1} - 1}{4(3(\Phi+1))^{2/\gamma}}}_{=:\Gamma} T\right)$$

We are left to bound

$$
\begin{aligned}
\sum_{i=0}^{I-1} K^{(i)} = \sum_{i=0}^{I-1} \lceil 2^{i\gamma} \rceil &\ge \sum_{i=0}^{I-1} 2^{i\gamma} = \frac{2^{\gamma I} - 1}{2^\gamma - 1} \\
&\ge \frac{2^{\gamma/1+\gamma \lg(\Gamma T)} - 1}{2^\gamma} \\
&= \frac{(\Gamma T)^{\gamma/1+\gamma} - 1}{2^\gamma} \\
&= \frac{\left(\frac{2^{\gamma+1} - 1}{4(3(\Phi+1))^{2/\gamma}} T\right)^{\gamma/1+\gamma} - 1}{2^\gamma}
\end{aligned}
$$

using our lower bound for $I$. Substituting this and the upper bound for $I$ into Eq. (9),

$$
\begin{aligned}
\min_{t=0,\ldots,T-1}\|\nabla f(w_t)\|^2 &\leq \frac{6IL\Delta + \frac{2}{3}C^2 \cdot I(3(\Phi+1))^{-2}}{\sum_{i=0}^{I-1} K^{(i)}} \\
&\leq \left(\frac{1}{\gamma+1}\right)\lg\left(\frac{2^{\gamma+2}}{(3(\Phi+1))^{2/\gamma}}T\right)\frac{6L\Delta + \frac{2}{3}C^2(3(\Phi+1))^{-2}}{\frac{\left(\frac{2^{\gamma+1}-1}{4(3(\Phi+1))^{2/\gamma}}T\right)^{\gamma/1+\gamma}-1}{2^\gamma}} \\
&\leq 12\lg\left(\frac{16}{3(\Phi+1)}T\right)\frac{L\Delta + C^2(3(\Phi+1))^{-2}}{\left(\frac{2^{\gamma+1}-1}{4(3(\Phi+1))^{2/\gamma}}T\right)^{\gamma/1+\gamma}-1} \\
&\leq 12\lg\left(\frac{16}{3(\Phi+1)}T\right)\frac{L\Delta + C^2(3(\Phi+1))^{-2}}{T^{\gamma/1+\gamma}(12(\Phi+1)^2)^{-\gamma/1+\gamma}-1} \\
&= \mathcal{O}\left(\frac{\log(T)}{T^{\gamma/1+\gamma}}\right),
\end{aligned}
$$

which yields our convergence rate in the nonconvex setting. $\qquad\square$

### A.2.2 DIMINISHING STEP SIZE: STRONGLY-CONVEX CASE

**Theorem 4** (Diminishing step size: strongly-convex case). *Suppose $f$ is $\mu$-Polyak-Łojasiewicz (PL), and that our setup satisfies Assumptions 1 and 2 Let $\epsilon > 0$ be any target error, and $\kappa = L/\mu$ be the condition number of the problem. After $T$ iterations of SGD (Eq. (1)) with a diminishing step-size (Eq. (7)), we can obtain*

$$
f(w_T) - f^* \leq \left(\frac{e^{1/\gamma}-1}{2\lceil 12\kappa\rceil(3(\Phi+1))^{2/\gamma}}\right)^{-\gamma}\left(\Delta + \frac{4C^2}{\mu}\frac{(3(\Phi+1))^{-2}}{e(1-\rho)^{\lceil 12\kappa\rceil}}\right) = \mathcal{O}\left(\frac{1}{T^\gamma}\right).
$$

*This implies the number of gradient evaluations to achieve $f(w_T) - f^* \leq \epsilon^2$ is*

$$
T = \mathcal{O}(\epsilon^{-2/\gamma}).
$$

*Proof.* We will begin with the same analysis technique as used in the non-convex case. To apply Lemma 1 to the iterations within a particular phase $i$, we need

$$
3\Phi \leq m^{\gamma/2} \qquad \text{and} \qquad \alpha \leq \frac{1}{6Lm}
$$

dropping the index $(i)$ for convenience and will re-introduce it later when appropriate. Lemma 1 gives us the following bound on the objective value between any length-$m$ number of iterations within that phase: for $\tau \in \{0, m-1, \ldots, (K-1)m-1\}$,

$$
\begin{aligned}
f(w_{\tau+m}) &\leq f(w_\tau) - \frac{\eta}{4}\|\nabla f(w_\tau)\|^2 + \frac{2C^2}{\eta}\alpha^2 m^{2-\gamma} \\
&\leq f(w_\tau) - \frac{\alpha m}{4}\|\nabla f(w_\tau)\|^2 + 2C^2\alpha m^{1-\gamma},
\end{aligned}
$$

since the step size is constant throughout one phase. Strong-convexity (or the Polyak-Łojasiewicz (PL) inequality) of $f$ implies $\|\nabla f(w)\|^2 \geq 2\mu(f(w)-f^*)$ for all $w \in \mathbb{R}^d$. Using this while subtracting $f^*$ on both sides leads to

$$
f(w_{\tau+m}) - f^* \leq \left(1 - \frac{\alpha m\mu}{2}\right)(f(w_\tau) - f^*) + 2C^2\alpha m^{1-\gamma}.
$$

Applying this recursively $K$ times gives

$$f(w_{mK}) - f^* \le \left(1 - \frac{\alpha m \mu}{2}\right)^K (f(w_0) - f^*) + 2C^2 \alpha m^{1-\gamma} \sum_{k=0}^{K-1} \left(1 - \frac{\alpha m \mu}{2}\right)^k$$

$$\le \left(1 - \frac{\alpha m \mu}{2}\right)^K (f(w_0) - f^*) + 2C^2 \alpha m^{1-\gamma} \sum_{k=0}^{\infty} \left(1 - \frac{\alpha m \mu}{2}\right)^k$$

$$= \left(1 - \frac{\mu}{12L}\right)^K (f(w_0) - f^*) + 2C^2 \alpha m^{1-\gamma} \frac{2}{\alpha m \mu}$$

$$= \left(1 - \frac{1}{12\kappa}\right)^K (f(w_0) - f^*) + 4C^2 m^{-\gamma} \mu^{-1},$$

where we have used

$$\alpha \le \frac{1}{6Lm} \implies \frac{\alpha m \mu}{2} \le \frac{\mu}{12L} < 1 \qquad \text{as } \mu \le L.$$

We now restore the phase index. Letting $T_i := \sum_{i'=0}^{i-1} m^{(i')} K^{(i')}$ be the total number of iterations passed after $i$ phases so that $T = T_I$, and $\Delta_t := f(w_t) - f^*$,

$$f(w_{T_{i+1}}) - f^* \le \left(1 - \frac{1}{12\kappa}\right)^{K^{(i)}} \Delta_{T_i} + \frac{4C^2}{\mu} (m^{(i)})^{-\gamma}.$$

Now let us choose

$$m^{(i)} = \lceil (3(\Phi + 1))^{2/\gamma} \cdot e^{i/\gamma} \rceil$$

$$K^{(i)} = K = \lceil 12\kappa \rceil = \lceil \frac{-1}{\log(1 - 1/12\kappa)} \rceil \ge 12.$$

Here the choice of $m^{(i)}$ guarantees that

$$(m^{(i)})^{\gamma/2} = \left(\lceil (3(\Phi + 1))^{2/\gamma} \cdot e^{i/\gamma} \rceil\right)^{\gamma/2} \ge \left((3(\Phi + 1))^{2/\gamma}\right)^{\gamma/2} e^{i/2} \ge 3\Phi.$$

Using the constant $K$ across all phases, our recursion can be simplified to

$$f(w_{T_{i+1}}) - f^* \le \left(1 - \frac{1}{12\kappa}\right)^K \Delta_{T_i} + \frac{4C^2}{\mu} (m^{(i)})^{-\gamma}.$$

Since this holds for any $i = 0, \ldots, I - 1$, applying recursion over $I$ phases yields,

$$f(w_T) - f^* \le (1 - \rho)^{KI} \Delta_0 + \frac{4C^2}{\mu} \sum_{i=0}^{I-1} (1 - \rho)^{Ki} (m^{(I-i)})^{-\gamma} \tag{10}$$

where we defined $\rho := 1/12\kappa$. Observe that for our choice of $m^{(i)}$ and $K$,

$$T = \sum_{i=0}^{I-1} K^{(i)} m^{(i)} = K \sum_{i=0}^{I-1} \lceil (3(\Phi + 1))^{2/\gamma} \cdot e^{i/\gamma} \rceil$$

$$\le K \sum_{i=0}^{I-1} 2(3(\Phi + 1))^{2/\gamma} \cdot e^{i/\gamma}$$

$$= 2K(3(\Phi + 1))^{2/\gamma} \sum_{i=0}^{I-1} e^{i/\gamma}$$

$$= \frac{2K(3(\Phi + 1))^{2/\gamma}}{e^{1/\gamma} - 1} (e^{I/\gamma} - 1) \le \frac{2K(3(\Phi + 1))^{2/\gamma}}{e^{1/\gamma} - 1} e^{I/\gamma},$$

which implies

$$I \ge \gamma \log\left(\frac{e^{1/\gamma} - 1}{2K(3(\Phi + 1))^{2/\gamma}} T\right),$$

and so

$$(1 - \rho)^{KI} \leq \exp(-\rho KI) \leq \exp\left(-\rho K\gamma \cdot \log\left(\frac{e^{1/\gamma} - 1}{2K(3(\Phi + 1))^{2/\gamma}}T\right)\right)$$

note that

$$\rho K\gamma = \gamma\rho\left\lceil\frac{-1}{\log(1 - \rho)}\right\rceil \geq \gamma.$$

This gives us

$$(1 - \rho)^{KI} \leq \left(\frac{e^{1/\gamma} - 1}{2K(3(\Phi + 1))^{2/\gamma}}\right)^{-\gamma} T^{-\gamma}. \tag{11}$$

Substituting this into Eq. (10),

$$f(w_T) - f^* \leq \Delta_0\left(\frac{e^{1/\gamma} - 1}{2K(3(\Phi + 1))^{2/\gamma}}\right)^{-\gamma} T^{-\gamma} + \frac{4C^2}{\mu}\sum_{i=0}^{I-1}(1 - \rho)^{Ki}(m^{(I-i)})^{-\gamma}. \tag{12}$$

For the last term,

$$\sum_{i=0}^{I-1}(1 - \rho)^{Ki}(m^{(I-i)})^{-\gamma} = \sum_{i=0}^{I-1}(1 - \rho)^{Ki}\lceil(3(\Phi + 1))^{2/\gamma} \cdot e^{I-i/\gamma}\rceil^{-\gamma}$$

$$\leq (3(\Phi + 1))^{-2}\sum_{i=0}^{I-1}(1 - \rho)^{Ki}e^{i-I}$$

$$= (3(\Phi + 1))^{-2}e^{-I}\sum_{i=0}^{I-1}(e(1 - \rho)^K)^i$$

$$= (3(\Phi + 1))^{-2}e^{-I}\frac{(e(1 - \rho)^K)^I}{e(1 - \rho)^K}$$

$$= (3(\Phi + 1))^{-2}\frac{(1 - \rho)^{KI}}{e(1 - \rho)^K}.$$

Substituting these into Eq. (12), and using Eq. (11) to bound $(1 - \rho)^{KI}$ gives us the convergence rate for strongly-convex functions using a diminishing step size:

$$f(w_T) - f^* \leq \Delta(1 - \rho)^{KI} + \frac{4C^2}{\mu}(3(\Phi + 1))^{-2}\frac{(1 - \rho)^{KI}}{e(1 - \rho)^K}$$

$$\leq (1 - \rho)^{KI}\left(\Delta + \frac{4C^2}{\mu}\right)\frac{(3(\Phi + 1))^{-2}}{e(1 - \rho)^K}$$

$$\leq \left(\frac{e^{1/\gamma} - 1}{2\lceil 12\kappa\rceil(3(\Phi + 1))^{2/\gamma}}\right)^{-\gamma}\left(\Delta + \frac{4C^2}{\mu}\frac{(3(\Phi + 1))^{-2}}{e(1 - \rho)^{\lceil 12\kappa\rceil}}\right)T^{-\gamma}$$

$$= \mathcal{O}\left(\frac{1}{T^\gamma}\right).$$

$\square$

## A.3 PROOF FOR LEMMA 1

We now prove the lemma that bounds the evolution of SGD over a length-$m$ analysis phase. This lemma is the key to our convergence analyses in both the diminishing and constant step size regimes. Before we proceed, we first state and prove the following bound on the gradient error when scaled by a nonincreasing sequence.

**Lemma 2.** *Suppose Assumption 2 holds. If $\{\rho_t\}$ is a deterministic, nonincreasing, and nonnegative sequence, then*

$$\left\|\sum_{t=\tau}^{\tau+m-1} \rho_t \left(\nabla f(w_\tau; x_t) - \nabla f(w_\tau)\right)\right\|^2 \leq \rho_\tau^2 \cdot m^{2-\gamma} \cdot \left(C^2 + \Phi^2 \|\nabla f(w_\tau)\|^2\right).$$

*Proof.* Let $\beta_t = \rho_t - \rho_{t+1}$ for $t \in \{\tau, \tau+1, \ldots, \tau+m-2\}$, and let $\beta_{\tau+m-1} = \rho_{t+m-1}$. Observe that these are all nonnegative, and

$$\left\|\sum_{t=\tau}^{\tau+m-1} \rho_t \left(\nabla f(w_\tau; x_t) - \nabla f(w_\tau)\right)\right\|^2$$

$$= \left\|\sum_{t=\tau}^{\tau+m-1} \sum_{k=t}^{\tau+m-1} \beta_k \left(\nabla f(w_\tau; x_t) - \nabla f(w_\tau)\right)\right\|^2$$

$$= \left\|\sum_{k=\tau}^{\tau+m-1} \sum_{t=\tau}^{k} \beta_k \left(\nabla f(w_\tau; x_t) - \nabla f(w_\tau)\right)\right\|^2$$

$$= \rho_\tau^2 \left\|\sum_{k=\tau}^{\tau+m-1} \frac{\beta_k}{\rho_\tau} \sum_{t=\tau}^{k} \left(\nabla f(w_\tau; x_t) - \nabla f(w_\tau)\right)\right\|^2.$$

Applying Jensen's inequality using $\sum_{k=\tau}^{\tau+m-1} \beta_k = \rho_\tau$,

$$\left\|\sum_{t=\tau}^{\tau+m-1} \rho_t \left(\nabla f(w_\tau; x_t) - \nabla f(w_\tau)\right)\right\|^2$$

$$\leq \rho_\tau^2 \sum_{k=\tau}^{\tau+m-1} \frac{\beta_k}{\rho_\tau} \left\|\sum_{t=\tau}^{k} \left(\nabla f(w_\tau; x_t) - \nabla f(w_\tau)\right)\right\|^2$$

$$= \rho_\tau^2 \sum_{k=\tau}^{\tau+m-1} \frac{\beta_k}{\rho_\tau} (k - \tau + 1)^2 \left\|\frac{1}{k - \tau + 1} \sum_{t=\tau}^{k} \left(\nabla f(w_\tau; x_t) - \nabla f(w_\tau)\right)\right\|^2.$$

Applying Assumption 2 on the squared norm,

$$\left\|\sum_{t=\tau}^{\tau+m-1} \rho_t \left(\nabla f(w_\tau; x_t) - \nabla f(w_\tau)\right)\right\|^2$$

$$\leq \rho_\tau^2 \sum_{k=\tau}^{\tau+m-1} \frac{\beta_k}{\rho_\tau} (k - \tau + 1)^2 \cdot \frac{1}{(k - \tau + 1)^\gamma} \left(C^2 + \Phi^2 \|\nabla f(w_\tau)\|^2\right)$$

$$= \rho_\tau^2 \sum_{k=\tau}^{\tau+m-1} \frac{\beta_k}{\rho_\tau} \cdot (k - \tau + 1)^{2-\gamma} \cdot \left(C^2 + \Phi^2 \|\nabla f(w_\tau)\|^2\right)$$

$$\leq \rho_\tau^2 \sum_{k=\tau}^{\tau+m-1} \frac{\beta_k}{\rho_\tau} \cdot m^{2-\gamma} \cdot \left(C^2 + \Phi^2 \|\nabla f(w_\tau)\|^2\right)$$

$$= \rho_\tau^2 \cdot m^{2-\gamma} \cdot \left(C^2 + \Phi^2 \|\nabla f(w_\tau)\|^2\right),$$

which is what we wanted to show. $\qquad \square$

We now recall the statement for Lemma 1. In the main paper, we state the lemma with constant step size. In the following proof, we prove it holds for non-decreasing step size, where constant step size will hold naturally as a special case. For this, we define $\eta := \sum_{t=\tau}^{\tau+m-1} \alpha_t$.

**Lemma 1.** *Suppose our setup satisfies Assumptions 1 and 2 and that we use a constant step size $\alpha$ For all timesteps $\tau \geq 0$, let $m > 0$ be some integer such that $3\alpha_\tau m^{1-\gamma/2}\Phi \leq \eta \leq \frac{1}{6L}$. Then the objective at timestep $\tau + m$ is bounded by*

$$f(w_{\tau+m}) \leq f(w_\tau) - \frac{\eta}{4}\|\nabla f(w_\tau)\|^2 + \frac{2}{\eta}\alpha_\tau^2 m^{2-\gamma}C^2.$$

*Proof.* Let $\eta$ and $g$ denote

$$\eta := \sum_{t=\tau}^{\tau+m-1} \alpha_t \qquad \text{and} \qquad g := \frac{1}{\eta}\sum_{t=\tau}^{\tau+m-1} \alpha_t \nabla f(w_t; x_t).$$

This means that $w_{\tau+m} = w_\tau - \eta g$. From Assumption 1 ($L$-Smoothness),

$$f(w_{\tau+m}) = f(w_\tau - \eta g)$$

$$\leq f(w_\tau) - \eta \langle \nabla f(w_\tau), g \rangle + \frac{\eta^2 L}{2}\|g\|^2$$

$$= f(w_\tau) - \frac{\eta}{2}\|\nabla f(w_\tau)\|^2 - \frac{\eta}{2}\|g\|^2 + \frac{\eta}{2}\|g - \nabla f(w_\tau)\|^2 + \frac{\eta^2 L}{2}\|g\|^2$$

$$\leq f(w_\tau) - \frac{\eta}{2}\|\nabla f(w_\tau)\|^2 + \frac{\eta}{2}\|g - \nabla f(w_\tau)\|^2,$$

where the last inequality follows because we assumed $\eta L \leq 1/6$ ($< 1$). Next, observe that

$$\frac{\eta}{2}\|g - \nabla f(w_\tau)\|^2$$

$$= \frac{1}{2\eta}\|\eta g - \eta \nabla f(w_\tau)\|^2$$

$$= \frac{1}{2\eta}\left\|\sum_{t=\tau}^{\tau+m-1} \alpha_t \left(\nabla f(w_t; x_t) - \nabla f(w_\tau)\right)\right\|^2$$

$$\leq \frac{1}{\eta}\left\|\sum_{t=\tau}^{\tau+m-1} \alpha_t \left(\nabla f(w_\tau; x_t) - \nabla f(w_\tau)\right)\right\|^2 + \frac{1}{\eta}\left\|\sum_{t=\tau}^{\tau+m-1} \alpha_t \left(\nabla f(w_t; x_t) - \nabla f(w_\tau; x_t)\right)\right\|^2.$$

To bound the second term,

$$\frac{1}{\eta}\left\|\sum_{t=\tau}^{\tau+m-1} \alpha_t \left(\nabla f(w_t; x_t) - \nabla f(w_\tau; x_t)\right)\right\|^2 = \eta\left\|\sum_{t=\tau}^{\tau+m-1} \frac{\alpha_t}{\eta} \left(\nabla f(w_t; x_t) - \nabla f(w_\tau; x_t)\right)\right\|^2$$

$$\leq \eta \sum_{t=\tau}^{\tau+m-1} \frac{\alpha_t}{\eta}\|(\nabla f(w_t; x_t) - \nabla f(w_\tau; x_t))\|^2$$

$$\leq \sum_{t=\tau}^{\tau+m-1} \alpha_t L^2\|w_t - w_\tau\|^2.$$

Combining the above,

$$\frac{\eta}{2}\|g - \nabla f(w_\tau)\|^2 \leq \frac{1}{\eta}\left\|\sum_{t=\tau}^{\tau+m-1} \alpha_t \left(\nabla f(w_\tau; x_t) - \nabla f(w_\tau)\right)\right\|^2 + \sum_{t=\tau}^{\tau+m-1} \alpha_t L^2\|w_t - w_\tau\|^2. \quad (13)$$

Applying Lemma 2 on the first term in Eq. (13) gives

$$\frac{1}{\eta}\left\|\sum_{t=\tau}^{\tau+m-1} \alpha_t \left(\nabla f(w_\tau; x_t) - \nabla f(w_\tau)\right)\right\|^2 \leq \frac{1}{\eta}\alpha_\tau^2 m^{2-\gamma}\left(C^2 + \Phi^2\|\nabla f(w_\tau)\|^2\right).$$

For the second term in Eq. (13),

$$
\sum_{t=\tau}^{\tau+m-1} \alpha_t L^2 \|w_t - w_\tau\|^2 = \sum_{t=\tau}^{\tau+m-1} \alpha_t L^2 \left\| \sum_{u=\tau}^{t-1} \alpha_u \nabla f(w_u; x_u) \right\|^2
$$

$$
\leq 3 \sum_{t=\tau}^{\tau+m-1} \alpha_t L^2 \left\| \sum_{u=\tau}^{t-1} \alpha_u \left( \nabla f(w_u; x_u) - \nabla f(w_\tau; x_u) \right) \right\|^2
$$

$$
+ 3 \sum_{t=\tau}^{\tau+m-1} \alpha_t L^2 \left\| \sum_{u=\tau}^{t-1} \alpha_u \left( \nabla f(w_\tau; x_u) - \nabla f(w_\tau) \right) \right\|^2
$$

$$
+ 3 \sum_{t=\tau}^{\tau+m-1} \alpha_t L^2 \left\| \sum_{u=\tau}^{t-1} \alpha_u \nabla f(w_\tau) \right\|^2
$$

$$
\leq 3 \sum_{t=\tau}^{\tau+m-1} \alpha_t L^4 \left( \sum_{u=\tau}^{t-1} \alpha_u \right) \sum_{u=\tau}^{t-1} \alpha_u \|w_u - w_\tau\|^2
$$

$$
+ 3 \sum_{t=\tau}^{\tau+m-1} \alpha_t L^2 \cdot \alpha_\tau^2 m^{2-\gamma} \left( C^2 + \Phi^2 \|\nabla f(w_\tau)\|^2 \right)
$$

$$
+ 3 \sum_{t=\tau}^{\tau+m-1} \alpha_t L^2 \left( \sum_{u=\tau}^{t-1} \alpha_u \right)^2 \|\nabla f(w_\tau)\|^2,
$$

where the second term in the last inequality follows from Lemma 2. Continuing,

$$
\sum_{t=\tau}^{\tau+m-1} \alpha_t L^2 \|w_t - w_\tau\|^2 \leq 3 \sum_{t=\tau}^{\tau+m-1} \alpha_t L^4 \eta \sum_{u=\tau}^{t-1} \alpha_u \|w_u - w_\tau\|^2
$$

$$
+ 3 \sum_{t=\tau}^{\tau+m-1} \alpha_t L^2 \alpha_\tau^2 m^{2-\gamma} \left( C^2 + \Phi^2 \|\nabla f(w_\tau)\|^2 \right)
$$

$$
+ 3\eta^3 L^2 \|\nabla f(w_\tau)\|^2
$$

$$
= 3\eta^2 L^2 \sum_{t=\tau}^{\tau+m-1} \alpha_t L^2 \|w_t - w_\tau\|^2
$$

$$
+ 3L^2 \eta \alpha_\tau^2 m^{2-\gamma} \left( C^2 + \Phi^2 \|\nabla f(w_\tau)\|^2 \right)
$$

$$
+ 3\eta^3 L^2 \|\nabla f(w_\tau)\|^2.
$$

This implies that

$$
(1 - 3\eta^2 L^2) \sum_{t=\tau}^{\tau+m-1} \alpha_t L^2 \|w_t - w_\tau\|^2 \leq 3L^2 C^2 \eta \alpha_\tau^2 m^{2-\gamma}
$$

$$
+ \left( 3\eta^3 L^2 + 3L^2 \eta \alpha_\tau^2 m^{2-\gamma} \Phi^2 \right) \|\nabla f(w_\tau)\|^2.
$$

And because by our assumption, $\alpha_\tau^2 m^{2-\gamma} \Phi^2 \leq \frac{\eta^2}{9}$, it follows that

$$
(1 - 3\eta^2 L^2) \sum_{t=\tau}^{\tau+m-1} \alpha_t L^2 \|w_t - w_\tau\|^2 \leq 3L^2 C^2 \eta \alpha_\tau^2 m^{2-\gamma} + \frac{10}{3} \eta^3 L^2 \|\nabla f(w_\tau)\|^2.
$$

Since we also assumed $\eta \leq \frac{1}{6L}$, it follows that $3\eta^2 L^2 \leq \frac{1}{12}$, and so

$$
\left( 1 - \frac{1}{12} \right) \sum_{t=\tau}^{\tau+m-1} \alpha_t L^2 \|w_t - w_\tau\|^2 \leq 3L^2 C^2 \eta \alpha_\tau^2 m^{2-\gamma} + \frac{5}{54} \eta \|\nabla f(w_\tau)\|^2,
$$

which gives

$$\sum_{t=\tau}^{\tau+m-1} \alpha_t L^2 \|w_t - w_\tau\|^2 \leq \frac{12}{11} \cdot 3L^2 C^2 \eta \alpha_\tau^2 m^{2-\gamma} + \frac{12}{11} \cdot \frac{5}{54} \eta \|\nabla f(w_\tau)\|^2$$

$$= \frac{36}{11} L^2 C^2 \eta \alpha_\tau^2 m^{2-\gamma} + \frac{10}{99} \eta \|\nabla f(w_\tau)\|^2.$$

So, putting this all together,

$$f(w_{\tau+m}) \leq f(w_\tau) - \frac{\eta}{2}\|\nabla f(w_\tau)\|^2 + \frac{1}{\eta}\alpha_\tau^2 m^{2-\gamma}\left(C^2 + \Phi^2\|\nabla f(w_\tau)\|^2\right)$$

$$+ \frac{36}{11}L^2 C^2 \eta \alpha_\tau^2 m^{2-\gamma} + \frac{10}{99}\eta\|\nabla f(w_\tau)\|^2$$

$$= f(w_\tau) - \frac{\eta}{2}\|\nabla f(w_\tau)\|^2 + \frac{1}{\eta}\alpha_\tau^2 m^{2-\gamma}C^2$$

$$+ \frac{1}{\eta}\alpha_\tau^2 m^{2-\gamma}\Phi^2\|\nabla f(w_\tau)\|^2$$

$$+ \frac{36}{11}L^2 C^2 \eta \alpha_\tau^2 m^{2-\gamma} + \frac{10}{99}\eta\|\nabla f(w_\tau)\|^2$$

$$\leq f(w_\tau) - \frac{\eta}{2}\|\nabla f(w_\tau)\|^2 + \frac{1}{\eta}\alpha_\tau^2 m^{2-\gamma}C^2$$

$$+ \frac{1}{9}\eta\|\nabla f(w_\tau)\|^2$$

$$+ 4L^2 C^2 \eta \alpha_\tau^2 m^{2-\gamma} + \frac{10}{99}\eta\|\nabla f(w_\tau)\|^2$$

$$= f(w_\tau) - \frac{\eta}{2}\|\nabla f(w_\tau)\|^2 + \frac{1}{\eta}\alpha_\tau^2 m^{2-\gamma}C^2$$

$$+ 4L^2 C^2 \eta \alpha_\tau^2 m^{2-\gamma} + \frac{21}{99}\eta\|\nabla f(w_\tau)\|^2$$

$$\leq f(w_\tau) - \frac{\eta}{2}\|\nabla f(w_\tau)\|^2 + \frac{1}{\eta}\alpha_\tau^2 m^{2-\gamma}C^2$$

$$+ \frac{1}{4\eta}C^2\alpha_\tau^2 m^{2-\gamma} + \frac{1}{4}\eta\|\nabla f(w_\tau)\|^2$$

$$\leq f(w_\tau) - \frac{\eta}{4}\|\nabla f(w_\tau)\|^2 + \frac{2}{\eta}\alpha_\tau^2 m^{2-\gamma}C^2.$$

This is what we wanted to show. $\square$

## A.4 CONVERGENCE ANALYSIS: CONSTANT STEP SIZE

### A.4.1 CONSTANT STEP SIZE: NON-CONVEX CASE

**Theorem 1** (Non-convex case). *Suppose that our setup satisfies Assumptions 1 and 2, and let $\epsilon > 0$ be any target error. Using SGD (1) with a constant step size $\alpha = \frac{1}{6L}\left[(4C/\epsilon + 3\Phi)^{2/\gamma}\right]^{-1}$, the number of steps $T$ needed to achieve $\min_{t=0,\cdots,T-1}\|\nabla f(w_t)\|^2 \leq \epsilon^2$ is at most*

$$T = \left\lceil \tfrac{48L\Delta}{\epsilon^2} \right\rceil \cdot \left\lceil \left(\tfrac{4C}{\epsilon} + 3\Phi\right)^{2/\gamma}\right\rceil = \tilde{\mathcal{O}}\left(\tfrac{C^{2/\gamma}L\Delta}{\epsilon^{2+2/\gamma}} + \tfrac{\Phi^{2/\gamma}L\Delta}{\epsilon^2} + \tfrac{C^{2/\gamma}}{\epsilon^{2/\gamma}} + \Phi^{2/\gamma}\right).$$

*Proof.* To satisfy the requirements of Lemma 1 in the constant step size, constant $m$ case, we need

$$3\alpha m^{1-\gamma/2}\Phi \leq m\alpha \leq \frac{1}{6L}.$$

This breaks apart into first a constraint on only $m$:

$$3\Phi \leq m^{\gamma/2}, \tag{14}$$

and then a constraint on $\alpha$ in terms of $m$:

$$\alpha \leq \frac{1}{6Lm}.$$

In the nonconvex setting, we invoke Lemma 1 followed by summing up over $K$ phases and telescoping,

$$\frac{\alpha m}{4}\sum_{k=0}^{K-1}\|\nabla f(w_{mk})\|^2 \leq f(w_0) - f^* + \frac{2K}{\alpha m}\alpha^2 m^{2-\gamma}C^2$$

$$= f(w_0) - f^* + 2K\alpha m^{1-\gamma}C^2,$$

which gives a rate of

$$\frac{1}{K}\sum_{k=0}^{K-1}\|\nabla f(w_{mk})\|^2 \leq \frac{4(f(w_0) - f^*)}{\alpha mK} + 8m^{-\gamma}C^2.$$

If we want to minimize the right side, observe that we will always want to set $\alpha$ as large as possible, i.e. set $\alpha = 1/6Lm$. This gives

$$\frac{1}{K}\sum_{k=0}^{K-1}\|\nabla f(w_{mk})\|^2 \leq \frac{24L(f(w_0) - f^*)}{K} + 8m^{-\gamma}C^2.$$

Now, for this to all be less than $\epsilon^2$, it suffices for

$$\frac{24L(f(w_0) - f^*)}{K} \leq \frac{\epsilon^2}{2} \quad \text{and} \quad 8m^{-\gamma}C^2 \leq \frac{\epsilon^2}{2}.$$

The former occurs when

$$K \geq \frac{48L(f(w_0) - f^*)}{\epsilon^2},$$

while the latter happens when

$$\frac{4C}{\epsilon} \leq m^{\gamma/2}. \tag{15}$$

If we let $\Delta = f(w_0) - f^*$, using Eqs. (14) and (15) and taking the minimum $m$ and $K$ required, we can bound the number of iterations as

$$T = mK \geq \left\lceil \frac{48L\Delta}{\epsilon^2} \right\rceil \cdot \left\lceil \left(\frac{4C}{\epsilon} + 3\Phi\right)^{2/\gamma}\right\rceil.$$

So,

$$T = \mathcal{O}\left(\frac{C^{2/\gamma}L\Delta}{\epsilon^{2+2/\gamma}} + \frac{L\Delta\Phi^{2/\gamma}}{\epsilon^2} + \frac{C^{2/\gamma}}{\epsilon^{2/\gamma}} + \Phi^{2/\gamma}\right),$$

where we suppose that $\gamma$ acts as a constant. $\square$

### A.4.2 Constant step size: strongly-convex case

**Theorem 2.** *Suppose that $f$ satisfies the $\mu$-PL condition and our setup satisfies Assumptions 1 and 2. Let $\epsilon > 0$ be any target error, and $\kappa = L/\mu$ be the condition number of the problem. Using SGD (1) with a constant step size $\alpha = \frac{1}{6L} \left\lceil (8C^2/(\mu\epsilon^2) + 9\Phi^2)^{1/\gamma} \right\rceil^{-1}$, the number of steps $T$ needed to guarantee $f(w_T) - f^* \le \epsilon^2$ is at most*

$$T = \left\lceil 12\kappa \log \left( \frac{2\Delta}{\epsilon^2} \right) \right\rceil \cdot \left\lceil \left( \frac{8C^2}{\mu\epsilon^2} + 9\Phi^2 \right)^{1/\gamma} \right\rceil = \tilde{\mathcal{O}} \left( \frac{C^{2/\gamma}\kappa}{\mu^{1/\gamma}\epsilon^{2/\gamma}} + \kappa\Phi^{2/\gamma} + \kappa \right).$$

*Proof.* The strongly convex setting has the same constraints on $m$ and $\alpha$ as the non-convex setting. Recall that $\mu$-strong convexity or $\mu$-PL of $f$ implies that for all $x$, $\|\nabla f(x)\|^2 \ge 2\mu(f(x) - f^*)$. Applying this to the result of Lemma 1 gives

$$f(w_{\tau+m}) \le f(w_\tau) - \frac{\alpha m \mu}{2} (f(w_\tau) - f^*) + 2\alpha m^{1-\gamma} C^2,$$

which is equivalent to

$$f(w_{\tau+m}) - f^* \le \left( 1 - \frac{\alpha m \mu}{2} \right) (f(w_\tau) - f^*) + 2\alpha m^{1-\gamma} C^2.$$

Applying this recursively over $K$ intervals of length $m$ starting from $\tau = 0$,

$$f(w_{mK}) - f^* \le \left( 1 - \frac{\alpha m \mu}{2} \right)^K (f(w_0) - f^*) + 2\alpha m^{1-\gamma} C^2 \sum_{k=0}^{K-1} \left( 1 - \frac{\alpha m \mu}{2} \right)^k$$

$$\le \exp \left( -\frac{\alpha m \mu K}{2} \right) (f(w_0) - f^*) + 2\alpha m^{1-\gamma} C^2 \sum_{k=0}^{\infty} \left( 1 - \frac{\alpha m \mu}{2} \right)^k$$

$$= \exp \left( -\frac{\alpha m \mu K}{2} \right) (f(w_0) - f^*) + 2\alpha m^{1-\gamma} C^2 \left( 1 - \left( 1 - \frac{\alpha m \mu}{2} \right) \right)^{-1}$$

$$= \exp \left( -\frac{\alpha m \mu K}{2} \right) (f(w_0) - f^*) + 2\alpha m^{1-\gamma} C^2 \cdot \frac{2}{\alpha m \mu}$$

$$= \exp \left( -\frac{\alpha m \mu K}{2} \right) (f(w_0) - f^*) + \frac{4C^2}{\mu} m^{-\gamma}.$$

As in the non-convex case, it best advantages us to set $\alpha$ as large as possible. Setting $\alpha = 1/6Lm$ gives

$$f(w_{mK}) - f^* \le \exp \left( -\frac{\mu K}{12L} \right) (f(w_0) - f^*) + \frac{4C^2}{\mu} m^{-\gamma}.$$

For this to be less than $\epsilon^2$, it suffices for each of these two terms to be less than $\epsilon^2/2$. In the latter case, we would need

$$\frac{8C^2}{\mu\epsilon^2} \le m^\gamma. \tag{16}$$

The former case holds when

$$K \ge \frac{12L}{\mu} \log \left( \frac{2\Delta}{\epsilon^2} \right) = 12\kappa \log \left( \frac{2\Delta}{\epsilon^2} \right),$$

where $\kappa = L/\mu$. Applying a ceiling, using the minimum requirement on $K$ and $m$ from Eq. (16) above and Eq. (14) required in Lemma 1, we can lower bound the total number of iterations as

$$T = mK \ge \left\lceil 12\kappa \log \left( \frac{2\Delta}{\epsilon^2} \right) \right\rceil \cdot \left\lceil \left( \frac{8C^2}{\mu\epsilon^2} + 9\Phi^2 \right)^{1/\gamma} \right\rceil.$$

Ignoring logarithmic terms gives

$$T = \tilde{\mathcal{O}} \left( \kappa \left( \frac{C^2}{\mu\epsilon^2} \right)^{1/\gamma} + \kappa\Phi^{2/\gamma} \right).$$

$\square$

### A.5 Justifications for Assumption 2 under various example orderings

#### A.5.1 Proof for Proposition 1 (Arbitrary permutation)

**Proposition 1.** *Let Assumptions 1 and 3 hold, and suppose that we are using a permutation-based method of sampling, that is, any method such that $x_{kn}, x_{kn+1}, \dots, x_{kn+n-1}$ is a permutation of $x^{(0)}, x^{(1)}, \dots, x^{(n-1)}$ for all epochs $k \geq 0$. Any such method satisfies Assumption 2 with $\gamma = 2$, $C^2 = n^2 A^2$ and $\Phi^2 = n^2 B^2$.*

*Proof.* This result follows trivially from Assumption 3. Consider an arbitrary sum of gradient errors going from $\tau$ to $\tau + m - 1$:

$$\sum_{t=\tau}^{\tau+m-1} \nabla f(w_\tau; x_{\sigma(t)}) - \nabla f(w_\tau),$$

where $\sigma(t)$ is the index into the training set given by the permutation $\sigma$ used at time step $t$. Since the interval $\{\tau, \tau+1, \dots, \tau+m-1\}$ is arbitrary, this $m$ can potentially be greater than $n$, the epoch size. We can split this interval up as follows. Let $\tau_1$ be the first epoch boundary in the interval, such that all $t$ going from $\tau$ to $\tau_1 - 1$ are within the same epoch as $w_\tau$, or else $\tau_1 = \tau + m$ if there is no epoch boundary in the interval. Let $\tau_2$ be the last epoch boundary in the interval, such that all $t$ going from $\tau_2$ to $\tau + m - 1$ are within a later epoch than $w_\tau$ (it may be the case that $\tau_1 = \tau_2$). Then the sum

$$\sum_{t=\tau_1}^{\tau_2-1} \nabla f(w_\tau; x_{\sigma(t)}) - \nabla f(\tau)$$

must be zero, since this interval goes over full epochs. This leaves us with

$$\left\| \sum_{t=\tau}^{\tau+m-1} \nabla f(w_\tau; x_{\sigma(t)}) - \nabla f(w_\tau) \right\|$$
$$\leq \left\| \sum_{t=\tau}^{\tau_1-1} \nabla f(w_\tau; x_{\sigma(t)}) - \nabla f(w_\tau) \right\| + \left\| \sum_{t=\tau_2}^{\tau+m-1} \nabla f(w_\tau; x_{\sigma(t)}) - \nabla f(w_\tau) \right\|$$
$$\leq n \left( A^2 + B^2 \|\nabla f(w_\tau)\|^2 \right)^{1/2}.$$

where we have used the fact that the intervals $\{\tau, \dots, \tau_1\}$ and $\{\tau_2, \tau+m-1\}$ are of length at most $\lfloor n/2 \rfloor$ (if such interval is of length greater than $\lfloor n/2 \rfloor$, then bounding its norm is equivalent to bounding the sums of the remaining terms, for which there are at most $\lfloor n/2 \rfloor$ of them). Therefore

$$\left\| \frac{1}{m} \sum_{t=\tau}^{\tau+m-1} \nabla f(w_\tau; x_{\sigma(t)}) - \nabla f(w_\tau) \right\|^2 \leq \frac{1}{m^2} n^2 \left( A^2 + B^2 \|\nabla f(w_\tau)\|^2 \right),$$

and so Assumption 2 is satisfied with $\gamma = 2$, $C^2 = n^2 A^2$, and $\Phi^2 = n^2 B^2$. $\qquad\square$

#### A.5.2 Proof for Proposition 2 (Shuffle once)

**Proposition 2.** *Suppose that we are using the shuffle once variant of SGD to learn over a region $\mathcal{B} \in \mathbb{R}^d$ of radius at most $R$, such that the iterates $w_t$ are guaranteed to remain within this region. Assume that for all $w \in \mathcal{B}$ and all examples $x$ in the training set of size $n$, Assumption 1 (L-Smoothness) and Assumption 3 hold. Then with probability at least $1 - p$, Assumption 2 holds with $\gamma = 2$, $C^2 = \tilde{\mathcal{O}}(dA^2(n + B^2))$, and $\Phi^2 = \tilde{\mathcal{O}}(ndB^2)$.[3]*

*Proof.* We begin by invoking Lemma 6 (Permuted vector Hoeffding inequality) with

$$X_{i,j} = \frac{\nabla f(w; x_{i,j}) - \nabla f(w)}{\left( A^2 + B^2 \|\nabla f(w)\|^2 \right)^{1/2}}$$

---

[3]The probability $p$ is only present in log terms in these expressions, so it does not appear in the $\tilde{\mathcal{O}}$.

where we use $x_{i,j}$ to denote the $j$-th example after time step $i$, for all $i \in \{1, \ldots m\}$, $j \in \{1, \ldots, n\}$, with $n \geq m > 0$. Note that here we are using a fixed $w \in \mathcal{B}$ that does not depend on the permutation $\sigma$ used in shuffle once. Clearly, $\sum_{j=1}^{n} X_{i,j} = 0$ for all $i$ due to the periodicity in shuffle once, and $\|X_{i,j}\| \leq 1$ by our assumption. Therefore the requirements of Lemma 6 are satisfied, which gives the following high probability bound for some $\epsilon > 0$ on any sequence drawn without replacement:

$$\mathbf{P}\left(\left\|\sum_{t=1}^{m} \nabla f(w; x_{\sigma(t)}) - \nabla f(w)\right\|^2 \geq \epsilon^2 \left(A^2 + B^2 \|\nabla f(w)\|^2\right)\right) \leq 2e^2 \exp\left(-\frac{\epsilon^2}{32m}\right).$$

Using the periodicity property of shuffle once, i.e. if $\sigma$ is the drawn permutation at the beginning of training then $\sigma(t + n) = \sigma(t)$, which allows us to arbitrarily shift the starting time step in the above:

$$\mathbf{P}\left(\left\|\sum_{t=\tau}^{\tau+m-1} \nabla f(w; x_{\sigma(t)}) - \nabla f(w)\right\|^2 \geq \epsilon^2 \left(A^2 + B^2 \|\nabla f(w)\|^2\right)\right) \leq 2e^2 \exp\left(-\frac{\epsilon^2}{32m}\right).$$

Consider an arbitrary interval of length $m$ out of the permutation. As all of the $n(n-1)/2$ intervals are equivalent to an interval of size at most $\lfloor n/2 \rfloor$, by the sum to 0 property, we only need to consider intervals with $m \leq n/2$. This gives us

$$\mathbf{P}\left(\left\|\sum_{t=\tau}^{\tau+m-1} \nabla f(w; x_{\sigma(t)}) - \nabla f(w)\right\|^2 \geq \epsilon^2 \left(A^2 + B^2 \|\nabla f(w)\|^2\right)\right) \leq 2e^2 \exp\left(-\frac{\epsilon^2}{16n}\right).$$

By a union bound across all intervals we have

$$\mathbf{P}\left(\exists \tau, m \in \mathbb{Z}, \ \left\|\sum_{t=\tau}^{\tau+m-1} \nabla f(w; x_{\sigma(t)}) - \nabla f(w)\right\|^2 \geq \epsilon^2 \left(A^2 + B^2 \|\nabla f(w)\|^2\right)\right)$$
$$\leq e^2 n^2 \exp\left(-\frac{\epsilon^2}{16n}\right).$$

Unfortunately, we cannot take a union bound over $\mathbb{R}^n$ or $\mathcal{B}$ (or even just over $\{w_0, \ldots, w_T\}$), since the set may contain iterates visited by shuffle once that depend on $\sigma$, and thus Lemma 6 may not hold. Instead, suppose that we cover the whole region $\mathcal{B}$ with balls of radius $\delta$. We will be able to do this with $(1 + 2R/\delta)^d$ balls by Lemma 7 ($\epsilon$-net lemma). So, if the centers of the balls form a set $\mathcal{W}$, then

$$\mathbf{P}\left(\exists \tau, m \in \mathbb{Z}, \ \hat{w} \in \mathcal{W}, \ \left\|\sum_{t=\tau}^{\tau+m-1} \nabla f(\hat{w}; x_{\sigma(t)}) - \nabla f(\hat{w})\right\|^2 \geq \epsilon^2 \left(A^2 + B^2 \|\nabla f(\hat{w})\|^2\right)\right)$$
$$\leq e^2 n^2 \left(1 + \frac{2R}{\delta}\right)^d \exp\left(-\frac{\epsilon^2}{16n}\right).$$

Note that the centers we use to cover the region is completely independent of the running algorithm, therefore the union bound over Lemma 6 can be applied. Next, consider some $w_\tau$ that is not necessarily the center of a ball. The function

$$\sum_{t=\tau}^{\tau+m-1} \nabla f(w_\tau; x_{\sigma(t)}) - \nabla f(w_\tau)$$

is $Ln$-Lipschitz continuous, because each of the components is $L$-Lipschitz, and we can sum up only at most $n/2$ of them. Let $\hat{w} \in \mathcal{W}$ be such that $\|w_\tau - \hat{w}\| \leq \delta$. Adding and subtracting $\nabla f(\hat{w}; \sigma_{(i)})$ and $\nabla f(\hat{w})$, applying triangle inequality followed by Lipschitz continuity, we have

$$\left\|\sum_{t=\tau}^{\tau+m-1} \nabla f(w_\tau; x_{\sigma(t)}) - \nabla f(w_\tau)\right\| \leq 2Ln\delta + \left\|\sum_{t=\tau}^{\tau+m-1} \nabla f(\hat{w}; x_{\sigma(t)}) - \nabla f(\hat{w})\right\|$$
$$\leq 2Ln\delta + \epsilon\sqrt{A^2 + B^2 \|\nabla f(\hat{w})\|^2}. \qquad \text{(w.h.p.)}$$

Adding and subtracting $\nabla f(w_\tau)$ from $\nabla f(\hat{w})$ and bound using $\delta$ and Lipschitz continuity again,

$$\left\| \sum_{t=\tau}^{\tau+m-1} \nabla f(w_\tau; x_{\sigma(t)}) - \nabla f(w_\tau) \right\| \leq 2Ln\delta + \epsilon\sqrt{A^2 + B^2 \|\nabla f(\hat{w} - \nabla f(w_\tau) + \nabla f(w_\tau))\|^2}$$

$$\leq 2Ln\delta + \epsilon\sqrt{A^2 + 2B^2L^2\delta^2 + 2B^2\|\nabla f(w_\tau)\|^2},$$

and consequently

$$\left\| \sum_{t=\tau}^{\tau+m-1} \nabla f(w_\tau; x_{\sigma(t)}) - \nabla f(w_\tau) \right\|^2 \leq 4L^2n^2\delta^2 + 2\epsilon^2\left(A^2 + 2B^2L^2\delta^2 + 2B^2\|\nabla f(w_\tau)\|^2\right).$$

Note that this inequality will fail to hold with probability at most

$$e^2 n^2 \left(1 + \frac{2R}{\delta}\right)^d \exp\left(-\frac{\epsilon^2}{16n}\right).$$

If we want this to fail with probability less than some $p$, it suffices to set

$$\epsilon^2 = 16n\left(\log\left(\frac{e^2n^2}{p}\right) + d\log\left(1 + \frac{2R}{\delta}\right)\right).$$

This gives

$$\left\| \sum_{t=\tau}^{\tau+m-1} \nabla f(w_\tau; x_{\sigma(t)}) - \nabla f(w_\tau) \right\|^2 \leq 4L^2n^2\delta^2$$

$$+ 32n \cdot \left(\log\left(\frac{e^2n^2}{p}\right) + d\log\left(1 + \frac{2R}{\delta}\right)\right) \cdot \left(A^2 + 2B^2L^2\delta^2 + 2B^2\|\nabla f(w_\tau)\|^2\right).$$

If we set $\delta$ such that $L^2 n\delta^2 = A^2$, then we can bound this with

$$\left\| \sum_{t=\tau}^{\tau+m-1} \nabla f(w_\tau; x_{\sigma(t)}) - \nabla f(w_\tau) \right\|^2 \leq 4nA^2$$

$$+ 32n \cdot \left(\log\left(\frac{e^2n^2}{p}\right) + d\log\left(1 + \frac{2RL\sqrt{n}}{A}\right)\right) \cdot \left(A^2 + \frac{2A^2B^2}{n} + 2B^2\|\nabla f(w_\tau)\|^2\right),$$

which gives

$$\left\| \frac{1}{m} \sum_{t=\tau}^{\tau+m-1} \nabla f(w_\tau; x_{\sigma(t)}) - \nabla f(w_\tau) \right\|^2$$

$$\leq \frac{32n}{m^2} \cdot \left(\log\left(\frac{e^3n^2}{p}\right) + d\log\left(1 + \frac{2RL\sqrt{n}}{A}\right)\right) \cdot \left(A^2 + \frac{2A^2B^2}{n} + 2B^2\|\nabla f(w_\tau)\|^2\right).$$

Hiding logarithmic terms involving $n, p, R, L, A$, the shuffle once setup satisfies the requirement of Assumption 2 that with probability at least $1 - p$,

$$\left\| \frac{1}{m} \sum_{t=\tau}^{\tau+m-1} \nabla f(w_\tau; x_{\sigma(t)}) - \nabla f(w_\tau) \right\|^2 = \tilde{\mathcal{O}}\left(\frac{nd}{m^2}\left(A^2 + \frac{2A^2B^2}{n} + 2B^2\|\nabla f(w_\tau)\|^2\right)\right)$$

$$= \tilde{\mathcal{O}}\left(\frac{1}{m^2}\left(ndA^2 + 2dA^2B^2 + 2ndB^2\|\nabla f(w_\tau)\|^2\right)\right)$$

$$= \tilde{\mathcal{O}}\left(\frac{1}{m^2}\left(dA^2(n + 2B^2) + 2ndB^2\|\nabla f(w_\tau)\|^2\right)\right),$$

and so the parameters are $\gamma = 2$, $C^2 = \tilde{\mathcal{O}}(dA^2(n + B^2))$, and $\Phi^2 = \tilde{\mathcal{O}}(ndB^2)$. $\qquad\square$

### A.5.3 PROOF FOR PROPOSITION 3 (RANDOM RESHUFFLING)

We first introduce a lemma that bounds the distance between iterates obtained from SGD with random permutation and that obtained from deterministic gradient descent, over one epoch.

**Lemma 3.** *Consider a single epoch of SGD with constant step size $\alpha > 0$, where the examples come from a random permutation over the training set. Let Assumption 1 (L-Smoothness) and the bounded gradient error Assumption 3 be satisfied. Without loss of generality, assume that the epoch starts at time $t = 0$ at $w_0$. Let the sequence $u_t$ be defined by $u_0 = w_0$ and*

$$u_{t+1} = u_t - \alpha \nabla f(u_t),$$

*while*

$$w_{t+1} = w_t - \alpha \nabla f(w_t; x_t)$$

*for some $x_t$ chosen from the permutation. With probability at least $(1 - \delta)$ it will hold that for all $T \in \{1, \ldots, n\}$, if we set*

$$\alpha \leq \min \left\{ \frac{1}{64e(e^2 + 1)BnL \log\left(\frac{2e^2n}{\delta}\right)}, \frac{1}{2nL} \right\},$$

*then*

$$\|w_T - u_T\|^2 \leq 128\alpha^2 n(e^2 + 1)^2 \cdot \log\left(\frac{2e^2n}{\delta}\right) \left(A^2 + B^2 e^2 \|\nabla f(w_T)\|^2\right).$$

*Proof.* Given our setup, observe that we can write the difference between these sequences as

$$
\begin{aligned}
w_{t+1} - u_{t+1} &= w_t - u_t - \alpha \left(\nabla f(w_t; x_t) - \nabla f(u_t)\right) \\
&= w_t - u_t - \alpha \left(\nabla f(w_t; x_t) - \nabla f(u_t; x_t)\right) - \alpha \left(\nabla f(u_t; x_t) - \nabla f(u_t)\right),
\end{aligned}
$$

such that summing this up and using $w_0 = u_0$ gives

$$w_T - u_T = -\alpha \sum_{t=0}^{T-1} \left(\nabla f(w_t; x_t) - \nabla f(u_t; x_t)\right) - \alpha \sum_{t=0}^{T-1} \left(\nabla f(u_t; x_t) - \nabla f(u_t)\right),$$

and so

$$
\begin{aligned}
\|w_T - u_T\| &\leq \alpha \sum_{t=0}^{T-1} \|\nabla f(w_t; x_t) - \nabla f(u_t; x_t)\| + \alpha \left\|\sum_{t=0}^{T-1} \left(\nabla f(u_t; x_t) - \nabla f(u_t)\right)\right\|. \\
&\leq \alpha L \sum_{t=0}^{T-1} \|w_t - u_t\| + \alpha \left\|\sum_{t=0}^{T-1} \left(\nabla f(u_t; x_t) - \nabla f(u_t)\right)\right\|.
\end{aligned}
$$

Recall the update rule of $u_t$, with Assumption 1 (L-Smoothness), we obtain

$$\|\nabla f(u_{t+1}) - \nabla f(u_t)\| \leq \alpha L \|\nabla f(u_t)\|.$$

By the reverse triangle inequality, this implies

$$
\begin{aligned}
\|\nabla f(u_t)\| &\leq (1 - \alpha L)^{-1} \|\nabla f(u_{t+1})\| \\
\|\nabla f(u_{t+1})\| &\leq (1 + \alpha L) \|\nabla f(u_t)\|,
\end{aligned}
$$

which further implies for any $T \in \{1, \cdots, n\}$ and $t \in \{0, \cdots, T - 1\}$,

$$
\begin{aligned}
\|\nabla f(u_t)\| &\leq (1 - \alpha L)^{-n} \|\nabla f(u_T)\| \\
&= \left(1 + \frac{\alpha L}{1 - \alpha L}\right)^n \|\nabla f(u_T)\| \\
&\leq \exp\left(\frac{\alpha L n}{1 - \alpha L}\right) \|\nabla f(u_T)\| \\
&\leq \exp\left(2\alpha L n\right) \|\nabla f(u_T)\| \\
&\leq e \|\nabla f(u_T)\|,
\end{aligned}
$$

while for any $t \in \{T+1, \cdots, n\}$,

$$
\begin{aligned}
\|\nabla f(u_t)\| \leq & (1+\alpha L)^n \|\nabla f(u_T)\| \\
\leq & e\|\nabla f(u_T)\|,
\end{aligned}
$$

where we apply the condition that $\alpha \leq 1/(2nL)$. For a given $T$, consider the following vector set

$$
X_j = \frac{\nabla f(u_j; x_j) - \nabla f(u_j)}{\sqrt{A^2 + B^2 e^2 \|\nabla f(u_T)\|}}, \exists j \in \{0, \cdots, n\},
$$

it can be easily verified that they sum to zero.

Now we apply Lemma 6 (Permuted vector Hoeffding inequality) , for any $\gamma \geq 0$,

$$
\mathbf{P}\left( \left\| \sum_{t=0}^{T-1} (\nabla f(u_t; x_t) - \nabla f(u_t)) \right\| \geq \gamma \sqrt{A^2 + B^2 e^2 \|\nabla f(u_T)\|^2} \right) \leq 2e^2 \exp\left( -\frac{\gamma^2}{32T} \right)
$$

$$
\leq 2e^2 \exp\left( -\frac{\gamma^2}{32n} \right)
$$

as $T \leq n$. Now, this holds for just one $T$. By a union bound,

$$
\mathbf{P}\left( \exists T \in \{1, \ldots, n\}, \left\| \sum_{t=0}^{T-1} (\nabla f(u_t; x_t) - \nabla f(u_t)) \right\| \geq \gamma Q_T \right) \leq 2e^2 n \exp\left( -\frac{\gamma^2}{32n} \right).
$$

where $Q_T = \sqrt{A^2 + B^2 e^2 \|\nabla f(u_T)\|^2}$. Now, if we set $\gamma$ such that

$$
2e^2 n \exp\left( -\frac{\gamma^2}{32n} \right) = \delta \quad \Rightarrow \quad \gamma^2 = 32n \log\left( \frac{2e^2 n}{\delta} \right),
$$

then we get that

$$
\mathbf{P}\left( \exists T \in \{1, \ldots, n\}, \left\| \sum_{t=0}^{T-1} (\nabla f(u_t; x_t) - \nabla f(u_t)) \right\| \geq \gamma Q_T \right) \leq \delta.
$$

In this case, we will have that with probability at least $(1 - \delta)$,

$$
\|w_T - u_T\| \leq \alpha L \sum_{t=0}^{T-1} \|w_t - u_t\| + \alpha \gamma Q_T.
$$

If we let $\rho_0 = 0$ and

$$
\rho_T = \alpha L \sum_{t=0}^{T-1} \rho_t + \alpha \gamma Q_T,
$$

then $\|w_T - u_T\| \leq \rho_T$. Here, if $T > 0$,

$$
\rho_{T+1} - \rho_T = \alpha L \rho_T + \alpha \gamma (Q_{T+1} - Q_T),
$$

on the other hand, obviously $\rho_1 = \alpha \gamma Q_1$,

$$
\begin{aligned}
\rho_T =& \alpha\gamma \sum_{k=0}^{T-1} (1+\alpha L)^{T-k-1}(Q_{k+1} - Q_k) \\
=& \alpha\gamma \left( \sum_{k=1}^{T} (1+\alpha L)^{T-k} Q_k - \sum_{k=0}^{T-1} (1+\alpha L)^{T-k-1} Q_k \right) \\
\leq& \alpha\gamma \left( (1+\alpha L) \sum_{k=0}^{T-1} (1+\alpha L)^{T-k-1} Q_k - \sum_{k=0}^{T-1} (1+\alpha L)^{T-k-1} Q_k + Q_T \right) \\
\leq& \alpha\gamma \left( \alpha L e^{1/2} \sum_{k=0}^{T-1} Q_k + Q_T \right) \\
\leq& \alpha\gamma(e^2 + 1)Q_T \\
=& \alpha\gamma(e^2 + 1)\sqrt{A^2 + B^2 e^2 \|\nabla f(w_T)\|^2} \\
\leq& \alpha\gamma(e^2 + 1)(A + Be\|\nabla f(w_T)\|).
\end{aligned}
$$

Put it back we obtain

$$
\begin{aligned}
\|w_T - u_T\| &\leq \alpha\gamma(e^2 + 1)\left(A + Be\|\nabla f(u_T)\|\right) \\
&\leq \alpha\gamma(e^2 + 1)\left(A + BeL\|w_T - u_T\| + Be\|\nabla f(w_T)\|\right),
\end{aligned}
$$

which gives

$$
(1 - \alpha\gamma(e^2 + 1)BeL)\|w_T - u_T\| \leq \alpha\gamma(e^2 + 1)\left(A + Be\|\nabla f(w_T)\|\right).
$$

If we require

$$
\alpha \leq \frac{1}{64e(e^2 + 1)BnL\log\left(\frac{2e^2 n}{\delta}\right)}
$$

Squaring and substituting the value of $\gamma$ gives

$$
\|w_T - u_T\|^2 \leq 128\alpha^2 n(e^2 + 1)^2 \cdot \log\left(\frac{2e^2 n}{\delta}\right)\left(A^2 + B^2 e^2\|\nabla f(w_T)\|^2\right)
$$

with probability at least $(1 - \delta)$. $\qquad\qquad\square$

We now provide justifications to Assumption 2 for the random reshuffling scheme.

**Proposition 3.** *Suppose that we are using the random reshuffling variant of SGD. Assume that for all $w \in \mathbb{R}^d$ and all examples, Assumption 1 (L-Smoothness) and Assumption 3 hold. For some $p \in (0, 1)$, set the constant step size to satisfy $\alpha \leq \left(\max\left\{1460 BnL \cdot \log\left(4e^2 T/p\right), 2nL\right\}\right)^{-1}$. Then with probability at least $1 - p$, Assumption 2 holds with $\gamma = 2$, $C^2 = \tilde{O}(nA^2)$ and $\Phi^2 = \tilde{O}(nB^2)$.*

*Proof.* For some $\gamma > 0$ (but different from the $\gamma$ of Lemma 3, consider the event that for some specific epoch $k$, for some $w_\tau \in \{w_0, w_1, \ldots, w_{n(k-1)}\} \cup \{u_0, u_1, \ldots, u_n\}$, where the $u_i$ are the $u$ from Lemma 3 for epoch $k$, and for some $\tau$ and $m_k$ such that $n(k-1) \leq \tau < \tau + m_k \leq nk$,

$$
\left\|\sum_{t=\tau}^{\tau+m_k-1} \nabla f(w_\tau; x_{\sigma(t)}) - \nabla f(w_\tau)\right\| \leq \gamma\sqrt{A^2 + B^2\|\nabla f(w_\tau)\|^2}.
$$

Since all the $x_{\sigma(t)}$ are from the $k$-th epoch, but $w_\tau$ is independent of any randomness in the $k$-th epoch (as it is either a point visited in a previous epoch, or a value from the $u$ sequence which depends only on the position at the start of the $k$-th epoch and not on any $k$-th epoch randomness), it follows that we can apply Lemma 6 (Permuted vector Hoeffding inequality) on either this sum or, alternatively, the terms from epoch $k$ but not in the sum (the terms left out) to get that

$$
\mathbf{P}\left(\left\|\sum_{t=\tau}^{\tau+m_k-1} \nabla f(w_\tau; x_{\sigma(t)}) - \nabla f(w_\tau)\right\| \geq \gamma\sqrt{A^2 + B^2\|\nabla f(w_\tau)\|^2}\right)
$$
$$
\leq 2e^2 \exp\left(-\frac{\gamma^2}{32\min(m_k, n - m_k)}\right).
$$

As $m_k \leq n$, it follows that

$$
\mathbf{P}\left(\left\|\sum_{t=\tau}^{\tau+m_k-1} \nabla f(w_\tau; x_{\sigma(t)}) - \nabla f(w_\tau)\right\| \geq \gamma\sqrt{A^2 + B^2\|\nabla f(w_\tau)\|^2}\right)
$$
$$
\leq 2e^2 \exp\left(-\frac{\gamma^2}{16n}\right).
$$

Now, by a union bound the probability that there exists some $\tau$, $w_\tau$, and $m_k$ such that the average gradient error is large is bounded by

$$
\mathbf{P}\left(\exists\left\|\sum_{t=\tau}^{\tau+m-1} \nabla f(w_\tau; x_{\sigma(t)}) - \nabla f(w)\right\| \geq \gamma\sqrt{A^2 + B^2\|\nabla f(w_\tau)\|^2}\right)
$$
$$
\leq 2e^2 nT^2 \exp\left(-\frac{\gamma^2}{16n}\right),
$$

where $T$ is the total number of iterations across all epochs (and we assume that we finish all epochs so $n$ divides $T$). This follows from the fact that there are at most $T$ such $\tau$ that we could take on, at most $T$ values of $w_\tau$ that can be adopted for each, and at most $n$ values $m_k$ can take on. If we set $\gamma$ such that

$$2e^2nT^2 \exp\left(-\frac{\gamma^2}{16n}\right) = \frac{p}{2} \quad \Rightarrow \quad \gamma^2 = 16n \log\left(\frac{4e^2nT^2}{p}\right), \tag{17}$$

then with probability at least $(1 - p/2)$, it will hold that for all epochs, $w_\tau$, $\tau$, and $m_k$,

$$\left\| \sum_{t=\tau}^{\tau+m_k-1} \nabla f(w_\tau; x_{\sigma(t)}) - \nabla f(w_\tau) \right\| \le \gamma\sqrt{A^2 + B^2\|\nabla f(w_\tau)\|^2}. \tag{18}$$

Additionally, set the $\delta$ in Lemma 3 to be $pn/2T$. By a union bound on the result of Lemma 3, across all $T/n$ epochs, with probability at least $(1 - p/2)$, it must follow that

$$\|w_T - u_T\|^2 \le 128\alpha^2 n(e^2+1)^2 \cdot \log\left(\frac{4e^2T}{p}\right)\left(A^2 + B^2e^2\|\nabla f(w_T)\|^2\right), \tag{19}$$

for the corresponding $u_t$ sequence for all epochs. Therefore, both of these inequalities Eqs. (18) and (19) hold together with probability at least $(1 - p)$.

Now, consider an arbitrary sum of gradient errors going from $\tau$ to $\tau + m - 1$. Note since the interval here is arbitrary, this $m$ can be different from $m_k$ and potentially greater than $n$. We can split this interval up as follows. Let $\tau_1$ be the first epoch boundary in the interval, such that all $t$ going from $\tau$ to $\tau_1 - 1$ are within the same epoch as $w_\tau$, or else $\tau_1 = \tau + m$ if there is no epoch boundary in the interval. Let $\tau_2$ be the last epoch boundary in the interval, such that all $t$ going from $\tau_2$ to $\tau + m - 1$ are within a later epoch than $w_\tau$ (it may be the case that $\tau_1 = \tau_2$). It follows that

$$\left\| \sum_{t=\tau}^{\tau+m-1} \nabla f(w_\tau; x_{\sigma(t)}) - \nabla f(w_\tau) \right\|$$
$$\le \left\| \sum_{t=\tau}^{\tau_1-1} \nabla f(w_\tau; x_{\sigma(t)}) - \nabla f(w_\tau) \right\|$$
$$+ \left\| \sum_{t=\tau_1}^{\tau_2-1} \nabla f(w_\tau; x_{\sigma(t)}) - \nabla f(w_\tau) \right\| + \left\| \sum_{t=\tau_2}^{\tau+m-1} \nabla f(w_\tau; x_{\sigma(t)}) - \nabla f(w_\tau) \right\|.$$

Observe that since the second of these sums must go over some number of full epochs, its value must be 0. Therefore,

$$\left\| \sum_{t=\tau}^{\tau+m-1} \nabla f(w_\tau; x_{\sigma(t)}) - \nabla f(w_\tau) \right\|$$
$$\le \left\| \sum_{t=\tau}^{\tau_1-1} \nabla f(w_\tau; x_{\sigma(t)}) - \nabla f(w_\tau) \right\| + \left\| \sum_{t=\tau_2}^{\tau+m-1} \nabla f(w_\tau; x_{\sigma(t)}) - \nabla f(w_\tau) \right\|.$$

Observe that the term in the first sum must be $nL/2$-Lipschitz continuous in $w$, because it can be written as the sum of at most $\lfloor n/2 \rfloor$ terms each of which is $L$-Lipschitz (either as the actual terms of the sum, or else the terms left out of the sum). So, add and subtract $\nabla f(u_\tau; x_{\sigma(t)})$ and $\nabla f(u_\tau)$,

$$\left\| \sum_{t=\tau}^{\tau+m-1} \nabla f(w_\tau; x_{\sigma(t)}) - \nabla f(w_\tau) \right\|$$
$$\le nL\|w_\tau - u_\tau\|$$
$$+ \left\| \sum_{t=\tau}^{\tau_1-1} \nabla f(u_\tau; x_{\sigma(t)}) - \nabla f(u_\tau) \right\| + \left\| \sum_{t=\tau_2}^{\tau+m-1} \nabla f(w_\tau; x_{\sigma(t)}) - \nabla f(w_\tau) \right\|.$$

Now applying Eq. (19) on the first term and Eq. (18) on the last two terms gives

$$
\left\| \sum_{t=\tau}^{\tau+m-1} \nabla f(w_\tau; x_{\sigma(t)}) - \nabla f(w_\tau) \right\|
$$
$$
\leq nL \cdot \sqrt{128\alpha^2 n(e^2+1)^2 \cdot \log\left(\frac{4e^2 T}{p}\right)\left(A^2 + B^2 e^2 \|\nabla f(w_\tau)\|^2\right)}
$$
$$
+ 2\gamma\sqrt{A^2 + B^2\|\nabla f(w_\tau)\|^2}.
$$

Squaring both sides for simplicity, we can bound this with

$$
\left\| \sum_{t=\tau}^{\tau+m-1} \nabla f(w_\tau; x_{\sigma(t)}) - \nabla f(w_\tau) \right\|^2
$$
$$
\leq 256\alpha^2 n^3 L^2 (e^2+1)^2 \cdot \log\left(\frac{4e^2 T}{p}\right)\left(A^2 + B^2 e^2 \|\nabla f(w_\tau)\|^2\right)
$$
$$
+ 8\gamma^2(A^2 + B^2\|\nabla f(w_\tau)\|^2),
$$

where we have also used Eq. (17) for $\gamma$. If we apply our requirement that $\alpha L n \leq 1/2$, we get

$$
\left\| \sum_{t=\tau}^{\tau+m-1} \nabla f(w_\tau; x_{\sigma(t)}) - \nabla f(w_\tau) \right\|^2 \leq 64n(e^2+1)^2 \cdot \log\left(\frac{4e^2 T}{p}\right)\left(A^2 + B^2 e^2 \|\nabla f(w_\tau)\|^2\right)
$$
$$
+ 128n(A^2 + B^2\|\nabla f(w_\tau)\|^2) \cdot \log\left(\frac{4e^2 nT^2}{p}\right).
$$

It follows that for random reshuffling with probability $1 - p$, if we set

$$
\alpha \leq \min\left\{ \frac{1}{64(e^2+1)BenL\log\left(\frac{4e^2 T}{p}\right)}, \frac{1}{2nL} \right\}
$$
$$
= \left[ \max\left\{ 1460BnL\log\left(\frac{4e^2 T}{p}\right), 2nL \right\} \right]^{-1},
$$

random reshuffling satisfies the requirement of Assumption 2 with $\gamma = 2$, $C^2 = \tilde{\mathcal{O}}(nA^2)$, and $\Phi^2 = \tilde{\mathcal{O}}(nB^2)$, where $\tilde{O}(\cdot)$ hides logarithmic terms in $n, T, p$. □

### A.5.4 PROOF FOR PROPOSITION 4 (RANDOM RESHUFFLING WITH DATA ECHOING)

The proof for Proposition 4 is nearly a repeat of the random reshuffling proof in Proposition 3. The trick is to re-define an epoch when data echoing is used. Instead of referring to an epoch as a random permutation of the $n$ examples in the training set as in vanilla random reshuffling, here we define the $cn$ samples as an epoch, where each example $\sigma(i)$ is repeated $c$ times. For instance, let $c = 3$, then the $k$-th epoch with permutation $\sigma_k$ is given by the sequence

$$
\dots, \underbrace{x_{\sigma_k(1)}, x_{\sigma_k(1)}, x_{\sigma_k(1)}, x_{\sigma_k(2)}, \dots, x_{\sigma_k(n-1)}, x_{\sigma_k(n)}, x_{\sigma_k(n)}, x_{\sigma_k(n)}}_{\text{examples used in epoch } k}, \dots
$$

As a result of this redefinition, Lemma 3 needs to be modified such that wherever $n$ appears, we now need $cn$. We omit the proof for the following lemma as it is a trivial adaptation from that of Lemma 3.

**Lemma 4.** *Consider a single epoch of SGD with constant step size $\alpha > 0$, where the examples come from a random permutation over the training set, each echoed $c$ times. Without loss of generality, assume that the epoch starts at time $t = 0$ at $w_0$. Let the sequence $u_t$ be defined by $u_0 = w_0$ and*

$$
u_{t+1} = u_t - \alpha\nabla f(u_t),
$$

*while*

$$w_{t+1} = w_t - \alpha \nabla f(w_t; x_t)$$

*for some $x_t$ chosen from the permutation. Under the same assumptions as Lemma 3, with probability at least $(1 - \delta)$ it will hold that for all $T \in \{1, \ldots, cn\}$,*

$$\|w_T - u_T\|^2 \leq 128\alpha^2 cn(e^2 + 1)^2 \cdot \log\left(\frac{2e^2 cn}{\delta}\right)\left(A^2 + B^2 e^2 \|\nabla f(w_T)\|^2\right).$$

The justifications for Assumption 2 for the random reshuffling with data echoing scheme can also be obtained from Appendix A.5.3 similarly. At appropriate places one should invoke Lemma 4 instead of Lemma 3, and replace $n$ with $cn$. We omit the proof as it is again a trivial modification.

### A.5.5   PROOF FOR PROPOSITION 5 (MARKOV CHAIN GRADIENT DESCENT (MCGD))

To justify Assumption 2 for MCGD, we will first need the following lemma.

**Lemma 5.** *Let $F$ be any vector-valued measurable function, and let $x_0, x_1, \ldots$ be a sequence of samples from a Markov chain with mixing time $t_{mix}$ and stationary distribution $\pi$ starting from an arbitrary initial distribution. If the function is constrained such that $\|F(x)\| \leq 1$ for all $x$, and if we also have $\mathbf{E}_{X \sim \pi}[F(X)] = 0$, then for any $\delta \in (0, 1)$,*

$$\mathbf{P}\left(\left\|\sum_{i=0}^{m-1} F(x_i)\right\| \geq 5t_{mix}\sqrt{2m \log\left(\frac{2e^2}{\delta}\right)}\right) \leq \delta,$$

*Proof.* Consider the Doob martingale

$$W_k = \mathbf{E}\left[\sum_{i=0}^{m-1} F(x_i) \mid \mathcal{F}_k\right],$$

where $\mathcal{F}_k$ contains all randomness up to timestep $k$, i.e. $x_0, x_1, \ldots, x_k$, and so (as usual for a Doob martingale) the martingale property is trivially satisfied using repeated conditioning:

$$\mathbf{E}\left[W_{k+1} \mid \mathcal{F}_{k+1}\right] = \mathbf{E}\left[\mathbf{E}\left[\sum_{i=0}^{m-1} F(x_i) \mid \mathcal{F}_{k+1}\right] \mid \mathcal{F}_k\right] = \mathbf{E}\left[\sum_{i=0}^{m-1} F(x_i) \mid \mathcal{F}_k\right] = W_k.$$

Observe that the sum we want is $W_m = \mathbf{E}\left[\sum_{i=0}^{m-1} F(x_i) \mid \mathcal{F}_m\right] = \sum_{i=0}^{m-1} F(x_i)$, and that this sum has increments

$$
\begin{aligned}
W_{k+1} - W_k &= \mathbf{E}\left[\sum_{i=0}^{m-1} F(x_i) \mid \mathcal{F}_{k+1}\right] - \mathbf{E}\left[\sum_{i=0}^{m-1} F(x_i) \mid \mathcal{F}_k\right] \\
&= \mathbf{E}\left[\sum_{i=0}^{k+1} F(x_i) \mid \mathcal{F}_{k+1}\right] + \mathbf{E}\left[\sum_{i=k+2}^{m-1} F(x_i) \mid \mathcal{F}_{k+1}\right] \\
&\qquad - \mathbf{E}\left[\sum_{i=0}^{k} F(x_i) \mid \mathcal{F}_k\right] - \mathbf{E}\left[\sum_{i=k+1}^{m-1} F(x_i) \mid \mathcal{F}_k\right] \\
&= \sum_{i=0}^{k+1} F(x_i) - \sum_{i=0}^{k} F(x_i) + \sum_{i=k+2}^{m-1} \mathbf{E}\left[F(x_i) \mid \mathcal{F}_{k+1}\right] - \sum_{i=k+1}^{m-1} \mathbf{E}\left[F(x_i) \mid \mathcal{F}_k\right] \\
&= F(x_{k+1}) + \sum_{i=k+2}^{m-1} \mathbf{E}\left[F(x_i) \mid \mathcal{F}_{k+1}\right] - \sum_{i=k+1}^{m-1} \mathbf{E}\left[F(x_i) \mid \mathcal{F}_k\right].
\end{aligned}
$$

Now, observe that since the mixing time of the Markov chain is $t_{\text{mix}}$, for any $i \geq k$, if $\mu$ denotes the distribution of $x_i$ conditioned on $\mathcal{F}_k$, then using results from Levin & Peres (2017, Section 4.5)

$$\|\mu - \pi\|_{TV} \leq 2^{-\lfloor (i-k)/t_{\text{mix}} \rfloor}.$$

It follows from this and the fact that $F$ is bounded that

$$\|\mathbf{E}\left[F(x_i) \mid \mathcal{F}_k\right]\| \leq 2^{-\lfloor (i-k)/t_{\mathrm{mix}}\rfloor}.$$

Therefore,

$$
\begin{aligned}
\left\| \sum_{i=k+2}^{m-1} \mathbf{E}\left[F(x_i) \mid \mathcal{F}_{k+1}\right] \right\| &\leq \sum_{i=k+2}^{m-1} \|\mathbf{E}\left[F(x_i) \mid \mathcal{F}_{k+1}\right]\| \\
&\leq \sum_{i=k+2}^{m-1} 2^{-\lfloor (i-k-1)/t_{\mathrm{mix}}\rfloor} \\
&\leq \sum_{i=k+2}^{\infty} 2^{-\lfloor (i-k-1)/t_{\mathrm{mix}}\rfloor} = 2t_{\mathrm{mix}} - 1,
\end{aligned}
$$

and similarly for the last term. It follows that

$$\|W_{k+1} - W_k\| \leq 4t_{\mathrm{mix}}.$$

Therefore, by the vector Azuma's inequality of Hayes (2005, Theorem 1.8), for any $a > 0$,

$$\mathbf{P}\left(\|W_m - W_0\| \geq 4t_{\mathrm{mix}}a\right) \leq 2e^2 \exp\left(-\frac{a^2}{2m}\right).$$

On the other hand, by the same reasoning as before,

$$
\begin{aligned}
\|W_0\| &= \left\| \mathbf{E}\left[ \sum_{i=0}^{m-1} F(x_i) \mid \mathcal{F}_0 \right] \right\| \\
&\leq \|F(x_0)\| + \left\| \mathbf{E}\left[ \sum_{i=1}^{m-1} F(x_i) \mid \mathcal{F}_0 \right] \right\| \\
&\leq 1 + 2t_{\mathrm{mix}} - 1 = 2t_{\mathrm{mix}}.
\end{aligned}
$$

So,

$$\mathbf{P}\left(\|W_m\| \geq 2t_{\mathrm{mix}} + 4t_{\mathrm{mix}}a\right) \leq 2e^2 \exp\left(-\frac{a^2}{2m}\right).$$

Now, setting $a$ such that

$$2e^2 \exp\left(-\frac{a^2}{2m}\right) = \delta \quad \Rightarrow \quad a^2 = 2m \log\left(\frac{2e^2}{\delta}\right),$$

and noting that this makes $a \geq 2$, we get

$$\mathbf{P}\left(\left\| \sum_{i=0}^{m-1} F(x_i) \right\| \geq 5t_{\mathrm{mix}}\sqrt{2m \log\left(\frac{2e^2}{\delta}\right)}\right) \leq \delta,$$

which is what we wanted to show. $\qquad\square$

The justification for Assumption 2 then follows straightforwardly.

**Proposition 5.** *Suppose that we use samples $x_t$ from a Markov chain with mixing time $t_{mix}$. Assume that for all $w \in \mathbb{R}$ and all examples $x_t$, Assumption 3 holds. Then with probability at least $1 - p$, Assumption 2 holds with $\gamma = 1$, $C^2 = \tilde{\mathcal{O}}(A^2 t_{mix}^2)$, and $\Phi^2 = \tilde{\mathcal{O}}(B^2 t_{mix}^2)$.*

*Proof.* Observe that we need Lemma 5 to hold for all subintervals of examples, of which there are only at most $T^2$. So, setting $\delta$ to be $p/T^2$ in Lemma 5, we can show that for all $\tau, m$, the probability

$$\left\| \sum_{t=\tau}^{\tau+m-1} \left(\nabla f(w_\tau; x_t) - \nabla f(w_\tau)\right) \right\|^2 \geq 25 \left(A^2 + B^2 \|\nabla f(w_\tau)\|^2\right) t_{\mathrm{mix}}^2 2m \log\left(\frac{2e^2 T^2}{p}\right)$$

is at least $1 - p$. Equivalently,

$$\left\|\frac{1}{m}\sum_{t=\tau}^{\tau+m-1}(\nabla f(w_\tau; x_t) - \nabla f(w_\tau))\right\|^2 \leq \frac{50}{m}t_{\text{mix}}^2\left(A^2 + B^2\|\nabla f(w_\tau)\|^2\right) \cdot \log\left(\frac{2e^2T^2}{p}\right).$$

It follows that MCGD satisfies the requirements of Assumption 2 with $\gamma = 1$,

$$C^2 = 50A^2t_{\text{mix}}^2\log\left(\frac{2e^2T^2}{p}\right) \quad \text{and} \quad \Phi^2 = 50B^2t_{\text{mix}}^2\log\left(\frac{2e^2T^2}{p}\right).$$

The big-$\tilde{\mathcal{O}}$ expressions in the proposition statement immediately follow. $\qquad\square$

### A.5.6 Justification for QMC-based data augmentation with random reshuffling

Before we begin with the proof, let us first present the introductory material necessary on quasi-Monte Carlo methods. Similar to Monte Carlo integration, QMC is also used for numerical integration but using low-discrepancy sequences instead of pseudorandom number sequences. Concretely, the problem is to approximate the integral of a function $f$ over some $s$-dimensional hypercube $[0, 1]^s$,

$$I(f) := \int_{[0,1]^s} f(x)dx$$

using the average of the function evaluated at a sequence of points $x_1, \ldots, x_m$,

$$I_m(f) := \frac{1}{m}\sum_{i=1}^m f(x_i).$$

The approximation error rate is defined as $\epsilon = |I(f) - I_m(f)|$, and it is well-known that in the case of Monte Carlo integration where the $x_i$'s are drawn uniformly at random from $[0, 1]^s$, the error rate is $\epsilon^2 = \mathcal{O}(1/m)$. If the sequence of $x_i$'s has low *star-discrepancy*, which is defined as

$$D_m^* := \sup_U\left|\frac{1}{m}\sum_{i=1}^m \mathbb{1}\{x_i \in U\} - \text{Volume}(U)\right|$$

where $U = \prod_{j=1}^s[0, b_i]$ for $b_i \in [0, 1)$, and the volume is measured using the $s$-dimensional Lebesgue measure. Intuitively, the smaller this quantity is the more evenly the sequence of points covers the space. Some popular low-discrepancy sequences include the Halton sequence, Sobol sequence, van der Corput sequence, etc., which are essentially deterministic sequences that are cleverly constructed to mimic random numbers but in fact have low star-discrepancy. For instance, the Halton sequence satisfies $D_m^* = \mathcal{O}((\log m)^s/m)$.

To ensure fast convergence of $I_m$ to $I$ as we increase $m$, in addition to using low discrepency sequences we also need $f$ to be relatively well-behaved. For this, the *Hardy-Krause variation* $V_{\text{HK}}$ is often used, for which we refer the reader to Drmota & Tichy (2006, Definition 1.13) for its detailed characterization. Most importantly, if $f$ has finite $V_{\text{HK}}$ on $[0, 1]^s$, then the *Koksma-Hlawka inequality* guarantees that

$$\epsilon \leq V_{\text{HK}}D_m^*.$$

This implies that using quasi-Monte Carlo with, for instance, the Halton sequence, to integrate a function $f$ with bounded Hardy-Krause variation, the error rate would be $\epsilon^2 = \mathcal{O}((\log m)^{2s}/m^2)$. In comparison to the Monte Carlo error rate, the QMC error rate can be much faster when the dimensionality $s$ is relatively small. For an in-depth exposition of the related materials we recommend Owen (2003) and Drmota & Tichy (2006).

We are now ready to restate our proposition from Section 5, in which we describe our QMC data augmentation setup.

**Proposition 6.** *Suppose that we are using the random reshuffling variant of SGD with QMC data augmentation as described. Assume that for all $w \in \mathbb{R}^d$ and all examples, Assumptions 1, 3, 4 and 5 hold for Equation 3. For some $p \in (0, 1)$, set the step size to be a constant such that $\alpha \leq \left(\max\{1460BnL \cdot \log(4e^2T/p), 2nL\}\right)^{-1}$. Then with probability at least $1 - p$, Assumption 2 holds with $\gamma = 2$, $C^2 = \tilde{\mathcal{O}}(n^2V^2\mathcal{C}_{QMC}^2\log(T)^{2s} + nA^2)$ and $\Phi^2 = \tilde{\mathcal{O}}(nB^2)$.*

*Proof.* Recall that the example used in the $t$-th iteration is being transformed as

$$x_t = \mathcal{A}(x^{(\sigma_{\lfloor t/n \rfloor}(t \bmod n))}, \zeta_{\lfloor t/n \rfloor + \sigma_{\lfloor t/n \rfloor}(t \bmod n)}).$$

We start from the average gradient error term, that is,

$$\left\| \frac{1}{m} \sum_{t=\tau}^{\tau+m-1} \nabla f(w_\tau; \mathcal{A}(x_t, \zeta_t)) - \nabla f(w_\tau) \right\|^2$$

$$= \left\| \frac{1}{m} \sum_{t=\tau}^{\tau+m-1} \nabla f(w_\tau; \mathcal{A}(x_t, \zeta_t)) - \frac{1}{n} \sum_{i=1}^{n} \mathop{\mathbf{E}}_{\zeta \sim [0,1]^s} \nabla f(w_\tau; \mathcal{A}(x^{(i)}; \zeta)) \right\|^2$$

$$\leq 2 \left\| \frac{1}{m} \sum_{t=\tau}^{\tau+m-1} \nabla f(w_\tau; \mathcal{A}(x_t, \zeta_t)) - \frac{1}{m} \sum_{t=\tau}^{\tau+m-1} \mathop{\mathbf{E}}_{\zeta \sim [0,1]^s} \nabla f(w_\tau; \mathcal{A}(x_t; \zeta)) \right\|^2$$

$$+ 2 \left\| \frac{1}{m} \sum_{t=\tau}^{\tau+m-1} \mathop{\mathbf{E}}_{\zeta \sim [0,1]^s} \nabla f(w_\tau; \mathcal{A}(x_t; \zeta)) - \frac{1}{n} \sum_{i=1}^{n} \mathop{\mathbf{E}}_{\zeta \sim [0,1]^s} \nabla f(w_\tau; \mathcal{A}(x^{(i)}; \zeta)) \right\|^2,$$

where the first step follows from the definition of $f$ in the data augmentation setup (Eq. (3)). In the second step, the first norm relates to the QMC approximation error while the second norm relates to the RR analysis. From the analysis in Section A.5.3 we know the second norm can be bounded asymptotically (and probabilistically) by $\tilde{\mathcal{O}}(nA^2) + \tilde{\mathcal{O}}(nB^2)\|\nabla f(w_\tau)\|^2/m^2$. Now we analyze the first term. Since we use a contiguous QMC subsequence on each individual example, for the period of $[\tau, \tau + m - 1]$, we define $\tau_j$ and $m_j$ as the starting point and length of example $x^{(j)}$ being chosen during this period, such that $\sum_{j=1}^{n} m_j = m$. With this notation, we can now rewrite the first norm as

$$\left\| \frac{1}{m} \sum_{t=\tau}^{\tau+m-1} \nabla f(w_\tau; \mathcal{A}(x_t, \zeta_t)) - \mathop{\mathbf{E}}_{\zeta \sim [0,1]^s} \nabla f(w_\tau; \mathcal{A}(x_t; \zeta)) \right\|^2$$

$$= \left\| \frac{1}{m} \sum_{j=1}^{n} \sum_{t=\tau_j}^{\tau_j+m_j-1} \nabla f\left(w_\tau; \mathcal{A}\left(x^{(j)}, \zeta_t\right)\right) - \mathop{\mathbf{E}}_{\zeta \sim [0,1]^s} \nabla f\left(w_\tau; \mathcal{A}\left(x^{(j)}; \zeta\right)\right) \right\|^2$$

$$\leq \frac{n}{m^2} \sum_{j=1}^{n} \left\| \sum_{t=\tau_j}^{\tau_j+m_j-1} \nabla f\left(w_\tau; \mathcal{A}\left(x^{(j)}, \zeta_t\right)\right) - \mathop{\mathbf{E}}_{\zeta \sim [0,1]^s} \nabla f\left(w_\tau; \mathcal{A}\left(x^{(j)}; \zeta\right)\right) \right\|^2$$

$$\leq \frac{2n(\tau_j+m_j)^2}{m^2} \sum_{j=1}^{n} \left\| \frac{1}{\tau_j+m_j} \sum_{t=0}^{\tau_j+m_j-1} \nabla f\left(w_\tau; \mathcal{A}\left(x^{(j)}, \zeta_t\right)\right) - \mathop{\mathbf{E}}_{\zeta \sim [0,1]^s} \nabla f\left(w_\tau; \mathcal{A}\left(x^{(j)}; \zeta\right)\right) \right\|^2$$

$$+ \frac{2n\tau_j^2}{m^2} \sum_{j=1}^{n} \left\| \frac{1}{\tau_j} \sum_{t=0}^{\tau_j-1} \nabla f\left(w_\tau; \mathcal{A}\left(x^{(j)}, \zeta_t\right)\right) - \mathop{\mathbf{E}}_{\zeta \sim [0,1]^s} \nabla f\left(w_\tau; \mathcal{A}\left(x^{(j)}; \zeta\right)\right) \right\|^2.$$

Now we can use the Koksma–Hlawka inequality (Aistleitner & Dick, 2014) to bound these norms, which gives us

$$\left\| \frac{1}{m} \sum_{t=\tau}^{\tau+m-1} \nabla f(w_\tau; \mathcal{A}(x_t, \zeta_t)) - \mathop{\mathbf{E}}_{\zeta \sim [0,1]^s} \nabla f(w_\tau; \mathcal{A}(x_t; \zeta)) \right\|^2$$

$$\leq \mathcal{O}\left[ \frac{n\mathcal{C}_{\text{QMC}}^2 V^2}{m^2} \sum_{j=1}^{n} \cdot \left(\log(\tau_j+m_j)^{2s} + \log(\tau_j)^{2s}\right) \right] \leq \mathcal{O}\left( \frac{n^2\mathcal{C}_{\text{QMC}}^2 V^2 \log(T)^{2s}}{m^2} \right).$$

Putting it together, we have shown that random reshuffling with QMC data augmentation satisfies Assumption 2 with $\gamma = 2$, $C^2 = \tilde{\mathcal{O}}\left(n^2\mathcal{C}_{\text{QMC}}^2 V^2 \log(T)^{2s} + nA^2\right)$, and $\Phi^2 = \tilde{\mathcal{O}}\left(nB^2\right)$. $\qquad\square$

### A.6 MISCELLANEOUS LEMMAS

We now collect some technical lemmas used in our analyses.

The following lemma is a Hoeffding-type concentration bound on the sums of random permutations of vectors. This lemma is particularly useful for simplifying the logic of the proofs of our shuffling propositions above, as it frees us from having to use Doob-martingale-type arguments throughout.

**Lemma 6** (Permuted vector Hoeffding inequality). *Let $n \geq m > 0$ be some integers, and let $X_{i,j}$ for $i \in \{1, \ldots, m\}$ and $j \in \{1, \ldots, n\}$ be vectors in $\mathbb{R}^d$. We also require for all $i, j$, $\|X_{i,j}\| \leq 1$, and that for all $i$,*

$$\sum_{j=1}^{n} X_{i,j} = 0.$$

*Then, if $\sigma : \{1, \ldots, n\} \to \{1, \ldots, n\}$ is a random permutation and that the individual $X_{i,j}$'s do not depend on $\sigma$,*

$$\mathbf{P}\left(\left\|\sum_{i=1}^{m} X_{i,\sigma(i)}\right\| \geq a\right) \leq 2e^2 \exp\left(-\frac{a^2}{32m}\right).$$

Note that this lemma trivially also applies to random subsamples of vector sums, by just letting $X_{i,j} = Y_j$ for the desired vector sequence $Y_j$.

*Proof.* This proof is adapted from the proof of Theorem 4.3 in Bercu et al. (2015). Their approach is more sophisticated and tends to a Bernstein-type inequality, but is specialized to only scalars. Consider sampling the elements of the permutation one at a time, and let $\mathcal{F}_j$ be the filtration containing the random variables $\sigma(1), \sigma(2), \ldots, \sigma(j)$ but not $\sigma(j+1), \ldots, \sigma_n$. For $k \leq n$, define the process

$$W_k = \mathbf{E}\left[\sum_{i=1}^{m} X_{i,\sigma(i)} \mid \mathcal{F}_k\right].$$

Observe that this must be a (vector) martingale process, as it is a Doob martingale. Explicitly, we can write it as

$$W_k = \mathbf{E}\left[\sum_{i=1}^{k} X_{i,\sigma(i)} + \frac{1}{n-k} \sum_{i=k+1}^{m} \sum_{j \notin \{\sigma(1),\ldots,\sigma(k)\}} X_{i,j} \mid \mathcal{F}_k\right]$$

$$= \mathbf{E}\left[\sum_{i=1}^{k} X_{i,\sigma(i)} - \frac{1}{n-k} \sum_{i=k+1}^{m} \sum_{j=1}^{k} X_{i,\sigma(j)} \mid \mathcal{F}_k\right],$$

where here we used the fact that the $X_{i,j}$'s sum to 0 over $j = [n]$. Thus, the increments of this process will have

$$W_k - W_{k-1} = \mathbf{E}\left[\sum_{i=1}^{k} X_{i,\sigma(i)} - \frac{1}{n-k} \sum_{i=k+1}^{m} \sum_{j=1}^{k} X_{i,\sigma(j)} \mid \mathcal{F}_k\right]$$

$$- \mathbf{E}\left[\sum_{i=1}^{k-1} X_{i,\sigma(i)} - \frac{1}{n-k+1} \sum_{i=k}^{m} \sum_{j=1}^{k-1} X_{i,\sigma(j)} \mid \mathcal{F}_{k-1}\right]$$

$$= X_{k,\sigma(k)} - \frac{1}{n-k} \mathbf{E}\left[\sum_{i=k+1}^{m} \sum_{j=1}^{k} X_{i,\sigma(j)} \mid \mathcal{F}_k\right]$$

$$+ \frac{1}{n-k+1} \mathbf{E}\left[\sum_{i=k}^{m} \sum_{j=1}^{k-1} X_{i,\sigma(j)} \mid \mathcal{F}_{k-1}\right]$$

$$= X_{k,\sigma(k)} - \frac{1}{n-k} \sum_{i=k+1}^{m} X_{i,\sigma(k)}$$

$$- \left(\frac{1}{n-k} - \frac{1}{n-k+1}\right) \sum_{i=k+1}^{m} \sum_{j=1}^{k-1} X_{i,\sigma(j)} + \frac{1}{n-k+1} \sum_{j=1}^{k-1} X_{k,\sigma(j)},$$

where we have repeatedly applied the martingale property. Now to bound all of these, first by our assumption

$$\left\|X_{k,\sigma(k)}\right\| \leq 1.$$

Also,

$$\frac{1}{n-k} \left\|\sum_{i=k+1}^{m} X_{i,\sigma(k)}\right\| \leq \frac{m-k}{n-k} \leq 1.$$

Next, again using the sum to 0 assumption, we have

$$\frac{1}{n-k+1} \left\|\sum_{j=1}^{k-1} X_{k,\sigma(j)}\right\| = \frac{1}{n-k+1} \left\|-\sum_{j=k}^{n} X_{k,\sigma(j)}\right\| \leq \frac{n-k+1}{n-k+1} = 1,$$

and finally, by a combination of these bounds,

$$\left(\frac{1}{n-k} - \frac{1}{n-k+1}\right) \left\|\sum_{i=k+1}^{m} \sum_{j=1}^{k-1} X_{i,\sigma(j)}\right\| = \frac{1}{(n-k)(n-k+1)} \left\|\sum_{i=k+1}^{m} \sum_{j=1}^{k-1} X_{i,\sigma(j)}\right\|$$

$$\leq \frac{1}{(n-k)(n-k+1)} \cdot (m-k) \cdot (n-k+1)$$

$$\leq 1.$$

It follows that the increment satisfies

$$\|W_k - W_{k-1}\| \leq 4$$

with probability 1. We can now apply the vector Azuma's inequality of Hayes (2005, Theorem 1.8). Applying this to $W_m$ gives the result that, for any $a$,

$$\mathbf{P}\left(\|W_m\| \geq a\right) \leq 2e^2 \exp\left(-\frac{a^2}{32m}\right).$$

This proves the lemma. □

The next lemma we state is a standard result showing that we can cover a region of radius $R$ with a bounded number of balls of radius $\epsilon$.

**Lemma 7** ($\epsilon$-net lemma). *For any region $\mathcal{D}$ of radius $R$ in d-dimensional space, and any $\epsilon > 0$, there exists a subset $\mathcal{S} \subseteq \mathcal{D}$ such that $\mathcal{S}$ is of size at most*

$$|\mathcal{S}| \leq \left(1 + \frac{2R}{\epsilon}\right)^d,$$

*and for every point $x \in \mathcal{D}$, there exists a $\hat{x} \in \mathcal{S}$ such that $\|x - \hat{x}\| \leq \epsilon$.*

*Proof.* Consider the following procedure. As long as there are points in $\mathcal{D}$ that are not within a distance $\epsilon$ of an existing point in $S$, choose one such point arbitrarily and add it to $S$. Observe that with this construction, any two points in $\mathcal{S}$ must be at a distance greater than $\epsilon$ from each other. This means that if we center a ball with radius $\epsilon/2$ at each of the points in $\mathcal{S}$, these balls will be disjoint.

The total volume of these balls will be $|\mathcal{S}| \cdot V_1 \cdot (\epsilon/2)^d$, where $V_1$ is the volume of the unit ball in $d$-dimensional space. However, the centers of all these balls must lie in $\mathcal{D}$, and so the balls themselves must all lie within a slightly larger region of radius $R + \epsilon/2$. This region will have volume $V_1 \cdot (R + \epsilon/2)^d$. This shows that our process must eventually stop, implying that $\mathcal{S}$ must exist, and gives us a bound on the size of $\mathcal{S}$ as

$$|\mathcal{S}| \leq \frac{(R + \epsilon/2)^d}{(\epsilon/2)^d} = \left(1 + \frac{2R}{\epsilon}\right)^d,$$

and the proof is complete. $\square$

