# OpenReview forum: "A General Analysis of Example-Selection for Stochastic Gradient Descent"
_ICLR.cc/2022/Conference — ICLR 2022 Spotlight_

### Official Review · Reviewer_SC9E · 2021-11-02

**Correctness:** 4
**Technical Novelty And Significance:** 3
**Empirical Novelty And Significance:** 3
**Recommendation:** 8
**Confidence:** 3

**Main Review:**

The paper derives an intuitive and simple result that SGD convergence depends on how well the average of consecutive example gradients approximate true gradient. Even though the result is simple, the authors show that it helps recover several previous convergence guarantees using their framework. Their framework also helps develop two new example selection algorithms.

**Summary Of The Paper:**

The paper studies the dependency of SGD convergence on order of examples. For smooth loss functions, the paper bounds SGD convergence based on averages of consecutive example gradients. Specifically better the average of consecutive example gradients approximate the objective gradient, faster the SGD convergence is. The paper recovers (and in some cases improves) the convergence results of various example selection schemes.

Based on the framework. the paper also introduces couple of new example selection sequences: QMC based data augmentation and greedy algorithm that optimizes on  average gradient metric. The authors also propose using stale gradients and random projection for faster approximate greedy example selection.

The paper evaluates their theoretical guarantees on several deep learning benchmarks and demonstrate the faster convergence with respect to other example selection techniques such as random sampling, shuffle once and random reshuffling.

**Summary Of The Review:**

The paper presents new albeit simple result that helps bound SGD convergence based on example sequence. The authors also propose two new selection algorithms and demonstrate their efficacy using experiments on MNIST and CIFAR.

---

> ### Author Response · Authors · 2021-11-15
> **Response to Reviewer SC9E**
>
> Thank you for your time and the positive feedback! Please let us know if you have any concerns.

---

### Official Review · Reviewer_nwty · 2021-11-03

**Correctness:** 4
**Technical Novelty And Significance:** 3
**Empirical Novelty And Significance:** 2
**Recommendation:** 8
**Confidence:** 4

**Main Review:**

The paper proposes a general metric which can be used to prove convergence rates for a range of stochastic optimization algorithms such as Random Reshuffling, Shuffle Once, SGD with replacement, or any technique that uses sequences of examples for computing the gradients. The metric quantifies how fast the mean of a sequence of $m$ gradients (evaluated at a fixed point) concentrates (as a function of $m$) around the true mean. Then, the paper proves theorems showing that the convergence rate for PŁ and non-convex objectives can be computed in terms of this metric. Hence, proving the convergence rate of any sampling scheme requires only computing this metric. Hence, this paper takes a step in the direction of unifying convergence analysis of sampling schemes.

Overall, the paper is well written, and presents new and intuitive ideas, and useful results.

I have the following major concerns:

1. The convergence rates in prior work for shuffle once do not have a dependence on dimension $d$, whereas Proposition 2 has such a dependence. This is a concern because modern machine learning applications have huge dimensions. Can this be removed? Is this fundamental to the approach proposed in this paper, or do the authors think this is just an artifact of the analysis? This seems to arise from the fact that the analysis tries to prove Assumption 2 on the entire space using an $\epsilon$-net argument. However, existing works only use something like Assumption 2 on the optimizer (Mishchenko et al. (2020) use Lemma 1 only on the optimizer $x_*$, and Ahn et al. (2020) use Lemma 5 in the proof of Lemma 26, only at the optimizer $0$). Maybe this approach can help?

2. Proposition 2 assumes that the iterates $w_t$ remain in a region of radius $R$. This does not seem justified. For convex objectives, we can imagine that the iterates do not move far away and are somehow attracted towards the minimizer, however for non-convex objectives, this intuition would not hold.

3. The paper says that Proposition 5 gives a faster rate than Sun et al. (2018). However, setting $q=1/2$ would give that $T=O(\epsilon^{-4})$ even for Sun et al.(2018). I think the authors should provide some more details in the comparison.

4. While comparing the results of Proposition 2 and 3 with Mishchenko et al. (2020) and Nguyen et al. (2020), the authors should also compare with the upper bounds provided in Ahn et al. (2020), as well as the lower bound provided in Safran and Shamir (2020) for SO and Rajput et al. (2020) for RR.

I also have the following (minor) suggestions:
1. Theorem 2 (along with Proposition 3) seems to suggest that RR beats SGD with replacement when $\epsilon = o(1/\sqrt{n})$, which would be when the number of epochs is more than $\kappa$. This seems to agree with the recent results by Safran and Shamir (2021) saying that RR beats SGD only after $\kappa$ epochs. A discussion on this would be very useful to the reader.

2. When comparing with older results, the paper assumes that $B=0$, which seems like a strong assumption. For example, if $f=\|w\|^2$, and one of the functions $f(w;x)=\frac{1}{2}\|w\|^2$, then this assumption is not satisfied. However, I think this might not be a major concern because even if $B$ is not 0, the comparison would remain similar.

References:

Mishchenko, Konstantin, Ahmed Khaled Ragab Bayoumi, and Peter Richtárik. "Random reshuffling: Simple analysis with vast improvements." Advances in Neural Information Processing Systems 33 (2020).

Ahn, Kwangjun, Chulhee Yun, and Suvrit Sra. "SGD with shuffling: optimal rates without component convexity and large epoch requirements." Advances in Neural Information Processing Systems 33 (2020).

Nguyen, L.M., Tran-Dinh, Q., Phan, D.T., Nguyen, P.H. and van Dijk, M., 2021. A unified convergence analysis for shuffling-type gradient methods. Journal of Machine Learning Research, 22(207), pp.1-44.

Sun, Tao, Yuejiao Sun, and Wotao Yin. "On Markov chain gradient descent." In Proceedings of the 32nd International Conference on Neural Information Processing Systems, pp. 9918-9927. 2018.

Safran, Itay, and Ohad Shamir. "Random Shuffling Beats SGD Only After Many Epochs on Ill-Conditioned Problems." arXiv preprint arXiv:2106.06880 (2021)

Safran, Itay, and Ohad Shamir. "How good is SGD with random shuffling?." In Conference on Learning Theory, pp. 3250-3284. PMLR, 2020.

Rajput, Shashank, Anant Gupta, and Dimitris Papailiopoulos. "Closing the convergence gap of SGD without replacement." In International Conference on Machine Learning, pp. 7964-7973. PMLR, 2020.


**Summary Of The Paper:**

The paper proposes a new condition (metric) on the sequence of samples seen by SGD (with or without replacement), which can be used to prove tight convergence rates for strongly convex and non-convex objectives. The metric quantifies how quickly the average gradient of a subsequence of length $m$ converges to the true gradient. This metric is computed for popular schemes like shuffle once, and random reshuffling, as well as for random reshuffling with echoing and Markov chain gradient descent. Then, the paper proposes to combine quasi-Monte Carlo based data augmentation with random reshuffling, proves the condition for it, and shows empirically that it performs well. The paper also proposes a technique which greedily tries to optimize the metric in the proposed condition.

**Summary Of The Review:**

This paper presents new and intuitive ideas, and useful results, and is a step in the direction of unifying convergence analysis for different sampling schemes of SGD.

---

> ### Author Response · Authors · 2021-11-15
> **Response to Reviewer nwty**
>
> We are glad that you are arguing in our favour and we greatly appreciate the
> constructive feedback! The major concerns regarding the dependence on $d$ and
> the bounded iterates assumption are also shared by Reviewer 2 (XLAG). We partially repeat
> them here with additional discussions specific to your suggestions.
>
> You make a good point and in fact, in the strongly convex setting,
> the iterates indeed remain bounded for any permutation-based SGD (see Lemma 1 of Rajput et al. (2021)).
> However, in the non-convex case or even the convex case, this is not necessarily true (one example would be logistic regression).
> One way for this bounded iterates to hold is to have a non-convex loss with bounded gradients with $\ell_2$ regularization,
> although we acknowledge that in deep learning bounded gradients are also often not satisfied.
>
> The dependence on $d$ can easily be removed (along with removing the bounded iterates assumption)
> by simply using Proposition 1 which holds for any permutation-based SGD, although in that case we
> would suffer an extra dependence on $n$ (instead of $\sqrt{n}$ in Proposition 2).
> As such, we view this as a trade-off between $n$ and $d$ as it certainly is the case that deep neural networks
> are often highly overparameterized, but the dataset size can also be huge.
>
> We believe the core contribution here is that we provide a versatile
> framework to prove accelerated rate for permutation-based SGD.
> Given any scan order, as long as we can prove that our Assumption 2 is satisfied with $\gamma=2$
> and appropriate constants, then we automatically have accelerated rates (compared to with replacement sampling)
> via our Theorems 1 and 2. The tools involving assumptions on the optimizer that you suggested
> are definitely helpful in proving convergence for a particular permutation-based ordering
> from scratch, where as our proof technique is to prove convergence rates under the general
> Assumption 2, followed by showing that it indeed holds for any window of example gradients.
>
> **References**
> * Rajput et al. (2021) Permutation-based SGD: is random optimal?
>
> > 1. The convergence rates in prior work for shuffle once do not have a dependence on dimension $d$ , whereas Proposition~2 has such a dependence. This is a concern because modern machine learning applications have huge dimensions. Can this be removed? Is this fundamental to the approach proposed in this paper, or do the authors think this is just an artifact of the analysis? This seems to arise from the fact that the analysis tries to prove Assumption 2 on the entire space using an $\epsilon$-net argument. However, existing works only use something like Assumption 2 on the optimizer (Mishchenko et al. (2020) use Lemma 1 only on the optimizer $w^*$ , and Ahn et al. (2020) use Lemma 5 in the proof of Lemma 26, only at the optimizer ). Maybe this approach can help?
>
> Please refer to the paragraphs above.
>
> > 2. *Proposition 2 assumes that the iterates $w_t$ remain in a region of radius $R$. This does not seem justified. For convex objectives, we can imagine that the iterates do not move far away and are somehow attracted towards the minimizer, however for non-convex objectives, this intuition would not hold.*
>
> Please refer to the paragraphs above.
>
>
> > 3. *The paper says that Proposition 5 gives a faster rate than Sun et al. (2018). However, setting $q=1/2$  would give that $T=O(\epsilon^{-4})$ even for Sun et al.(2018). I think the authors should provide some more details in the comparison.*
>
> The $q$ in Sun et al. (2018) can only be in $(1/2,1)$ and therefore our rate is strictly better.
>
> > 4. *Comparison to related works*
>
> We appreciate that you mentioned the work of Ahn et al. (2020),
> and the reason for not explicitly comparing against them is that they did not
> provide rates in the non-convex setting, and in the strongly convex setting
> a faster rate is achieved in Mishchenko et al. (2020). Nevertheless, your
> suggestion is appropriate and we should have compared the related results more thoroughly,
> which we will do in the revision.
>
> Same goes for the work of Safran and Shamir (2021), thanks for pointing out the agreement and we will definitely include a discussion in the revision.
>
> Finally, the case of $B=0$ is analogous to the bounded variance assumption typical in
> a non-convex SGD analysis, and the reason for considering this case
> when comparing to previous results is solely for simplicity.

---

### Official Review · Reviewer_XLAG · 2021-11-03

**Correctness:** 3
**Technical Novelty And Significance:** 3
**Empirical Novelty And Significance:** 3
**Recommendation:** 8
**Confidence:** 4

**Main Review:**

## Main
I am somewhere between "weak accept" and "accept" for this paper. On the one hand, I think the topic of this work is very promising and a small improvement in the convergence of RR may have a huge impact since it is widely used to train neural networks. Moreover, I am really glad to see that the authors managed to obtain guarantees for Shuffle Once that are better than the guarantees for Incremental Gradient established by Mishchenko et al. (2020), assuming dimension $d$ is smaller than $n$. On the other hand, many aspects of this work are worrisome. It is a bit disappointing to see that the rate in Proposition 2 depends on $d$ and that the proposition requires the domain to be bounded. But the main concern that I have is that all rates for RR and SO seem to be strictly worse than the rate of standard gradient descent. This is in contrast to the results of Theorem 1 of Mishchenko et al. (2020), which shows that RR and SO can be $n$ times faster than gradient descent as long as $\sigma^2$ is small and the loss functions are strongly convex. It seems that for the nonconvex setting there are no results showing that one can outperform gradient descent with RR, but I hoped that at least under some assumptions, there can be shown an improvement.

## Clarity
I think the presentation of the theoretical results could be improved. For instance, I would appreciate if the authors gave more details about Hardy-Krause variation bound and why we should expect it to be satisfied. I find the current discussion to be very limited and I believe the authors should include more information than just stating that they "are commonly used in analyzing QMC sequence". Similarly, I feel like I do not understand Assumption 5, which seems to be discussed neither in the main paper nor in the appendix. Finally, I think it would be nice to have a table that compares theoretical complexities with the prior literature.

## Experiments
The experiments look very insightful and I appreciate that the authors provided confidence intervals in addition to the average results. I am glad to see a numerical study of the average gradient error in Figure 1. It'd be very nice if the authors could study this phenomenon on a non-synthetic problem. The only concern that I have is about the results for ImageNet: Did the authors use a single random seed for it? If not, please also provide the standard errors in Table 1. In addition, I am a bit puzzled as to why the authors did not provide convergence plots for the ImageNet experiment?

## Please add discussion of limitations
Let me conclude by saying that I really appreciate the topic of the work, but I don't think this paper solves the direction. Many aspects can be improved and I encourage the authors to dedicate a paragraph or two in the conclusion section to describe potential improvements and the limitations of the current results.

### --MINOR ISSUES--
"optimizing an strongly convex" -> "optimizing a strongly convex"
"e.g." or "e.g.,": either put a comma after every "e.g." or do not put it at all
On page 5, in the "Shuffle once (SO)" section, the index is placed as superscript instead of subscript "$x_t = x^{(\sigma(t\ mod\ n))}$"
Cosine annealing would probably work better on both Cifar10 and Imagenet. In particular, training with weight decay=5e-4 with 300 epochs often allows to train Resnet20 to a test accuracy higher than 95%.
Theorem 3 and its proof use notation $lg()$. Is it a typo? I don't think this extra logarithmic notation is introduced anywhere.
"As all of the $n^2/2$ intervals are equivalent" did you mean $\frac{n(n-1)}{2}$ intervals? $n^2/2$ does not even make sense for $n$ odd.
In the proof of Theorem 3, when you say "By a union bound across all intervals we have", I think it should be the probability of *existence* of some $\tau$ and $m$ that the inequality holds, so $P(\exists \tau ...)$ instead of $P(\forall \tau ...)$, because you prove that this inequality may hold only rarely. The inequality itself, nevertheless, has the correct coefficients and can be proved by the union bound for the negation of the inequality inside $P()$. Similarly, when you define $\mathcal{W}$ later in the proof, the probability argument should have $\exists \hat w\in \mathcal{W}$.

**Summary Of The Paper:**

The paper studies the convergence of SGD, Random Reshuffling (RR), and, more generally, algorithms that process data examples in an order satisfying a certain assumption, which is a variant of bounded variance assumption with variance decreasing as $O(m^{-\gamma})$, where $\gamma$

**Summary Of The Review:**

This work revolves around a new assumption on the gradient variance that captures SGD, Random Reshulffing (RR), and Shuffle Once (SO), but it is not limited to these methods. As such, it is a bit more universal and it even allows the authors to obtain new rates for Shuffle Once in the nonconvex setting. Some parts of the theory are not clear to me, in particular, I do not feel like I understand Assumptions 4 and 5. I think some polishing of how the results are presented could help. On the other hand, I checked the proof of Proposition 2 and it seems quite novel to me. Mostly, the results look quite good and the theory is not a straightforward modification of prior work. It has some weaknesses though, in particular, it does not guarantee any benefit of RR or SO when comparing to plain gradient descent.
What I really like about this work is the new variants of data selection proposed by the authors. In particular, they show some promising results on ImageNet by using their new data augmentation technique and study its guarantees, although based on rather exotic assumptions. The authors also get insight from their main assumption to construct a better permutation ordering based on greedy selection. I think that this direction is very promising and may lead to improved training strategies in a wide range of applications.

---

> ### Author Response · Authors · 2021-11-15
> **Response to Reviewer XLAG (Part I)**
>
> Thank you for the detailed, constructive feedback and we are glad that
> you consider this work to be promising!
> Please see threads below where we attempt to address your concerns.
>
> > 1. *It is a bit disappointing to see that the rate in Proposition 2 depends on $d$ and that the proposition requires the domain to be bounded.*
>
> It is definitely reasonable to question the restrictiveness of the bounded iterates assumption.
> In the strongly convex setting, the iterates indeed remain bounded for any permutation-based SGD (see Lemma 1 of Rajput et al. (2021)).
> However, in the non-convex case or even the convex case, this is not necessarily true (one example would be logistic regression).
> One way for this bounded iterates to hold is to have a non-convex loss with bounded gradients with $\ell_2$ regularization,
> although we acknowledge that in deep learning bounded gradients are also often not satisfied.
>
> The dependence on $d$ can easily be removed (along with removing the bounded iterates assumption)
> by simply using Proposition 1 which holds for any permutation-based SGD, although in that case we
> would suffer an extra dependence on $n$ (instead of $\sqrt{n}$ in Proposition 2).
>
> **References**
> * Rajput et al. (2021) Permutation-based SGD: is random optimal?
>
> > 2. *But the main concern that I have is that all rates for RR and SO seem to be strictly worse than the rate of standard gradient descent. This is in contrast to the results of Theorem 1 of Mishchenko et al. (2020), which shows that RR and SO can be $n$ times faster than gradient descent as long as $\sigma^2$ is small and the loss functions are strongly convex. It seems that for the nonconvex setting there are no results showing that one can outperform gradient descent with RR, but I hoped that at least under some assumptions, there can be shown an improvement.*
>
> We agree that when comparing to GD, the rate from Theorem 1 of Mishchenko et al. (2020)
> can be $n$ times faster when $\sigma^2$ is small. However, as you pointed out, this result requires the assumption that
> the component functions are all strongly-convex, whereas we only assumed strong-convexity (or PL)
> on the overall objective. In fact, our strongly-convex analysis follows trivially from Lemma 1, which bounds the suboptimality
> of the objective rather than the iterates. Since this is a general lemma without any convexity assumptions,  this could be where the looseness in our bounds arises. However, as we are mainly interested in the non-convex setting
> and strived for a compact analysis framework, we did not specialize Lemma 1 to the convex or strongly-convex case.
>
>
> > 3. *Clarity:  For instance, I would appreciate if the authors gave more details about Hardy-Krause variation bound and why we should expect it to be satisfied. I find the current discussion to be very limited and I believe the authors should include more information than just stating that they "are commonly used in analyzing QMC sequence". Similarly, I feel like I do not understand Assumption 5, which seems to be discussed neither in the main paper nor in the appendix. Finally, I think it would be nice to have a table that compares theoretical complexities with the prior literature.*
>
> Thanks for pointing this out: we will include the introductory material
> necessary for quasi-Monte Carlo integration and the related theoretical tools
> to analyze its error bounds. Hopefully this will make Assumptions 4 and 5 more relatable.
>
> > 4. *Experiments: I am glad to see a numerical study of the average gradient error in Figure 1. It'd be very nice if the authors could study this phenomenon on a non-synthetic problem.*
>
> This is a great suggestion that we have neglected after running the synthetic experiments. However, to reproduce Figure 1b on a real dataset, we'll have to forgo the
> online setting as we no longer have access to an expression for the true gradient since
> the underlying distribution of the data would be unknown. Nonetheless, we will strive to include
> an offline version of Figure 1b using real data in the revised appendix.

---

> > ### Author Response · Authors · 2021-11-15
> > **Response to Reviewer XLAG (Part II)**
> >
> > > 5. *Experiments: The only concern that I have is about the results for ImageNet: Did the authors use a single random seed for it? If not, please also provide the standard errors in Table 1. In addition, I am a bit puzzled as to why the authors did not provide convergence plots for the ImageNet experiment?*
> >
> > In the ImageNet experiment, we log the best top1 validation accuracy for each algorithm among three runs, which generally follows https://github.com/pytorch/examples/blob/master/imagenet/main.py. Note that for ImageNet we did not train from scratch but finetuned the model from a pretrained checkpoint for 5 epochs as illustrated in the paper, since based on our findings from the CIFAR experiment (Figure 2), the effect of QMC would generally appear in the later stage of training. On the other hand, we agree with you that the convergence plots should also be included, we have added two figures on validation accuracy/loss in the Appendix A.1.1.
> >
> > > 6. *Other comments and minor issues*
> >
> > We really appreciate your effort on pointing out typos and sloppiness in the presentation,
> > we will make sure these are properly addressed.
> > In particular, the $\lg$ notation was used as a standard notation for $\log_2$
> > in computer science, although we recently discovered that $\lg$ is used to denote
> > $\log_{10}$ instead in mathematics. We will make the base $2$ clear in its context.
> > Your interpretation of the $n^2/2$ intervals is exactly correct --- we meant $n(n-1)/2$
> > or rather $O(n^2)$. Same goes for the quantifiers, we will correct these
> > and thank you for catching them!

---

> > > ### Comment · Reviewer_XLAG · 2021-11-29
> > > **Thank you for the feedback**
> > >
> > > > The dependence on $d$ can easily be removed (along with removing the bounded iterates assumption) by simply using Proposition 1 which holds for any permutation-based SGD, although in that case we would suffer an extra dependence on $n$ (instead of $\sqrt{n}$ in Proposition 2).
> > >
> > > I think that in this case the result wouldn't be better than the existing results on Incremental Gradient from prior literature.
> > >
> > > > Nonetheless, we will strive to include an offline version of Figure 1b using real data in the revised appendix.
> > >
> > > I can see this experiment in Appendix A, thank you for checking.
> > > The description of the experiment has some typos: "To measur the left hand side of Assumption 2" -> "To measure the left-hand side of Assumption 2"
> > >
> > > > On the other hand, we agree with you that the convergence plots should also be included, we have added two figures on validation accuracy/loss in the Appendix A.1.1.
> > >
> > > Thank you for providing the plots. I think many aspects of this experiment could be improved even further, for instance, testing the hypothesis on the improvements of the later stages, but since this is a theoretical work, I find it to be sufficient as proof of concept.
> > >
> > > One final comment, please change the representation of numbers from "7e-7" to $7\cdot 10^{-7}$: The former is confusing because you also use $e$ as the number, for instance, in the proof of Theorem 4.

---

### Official Review · Reviewer_rjuV · 2021-11-06

**Correctness:** 4
**Technical Novelty And Significance:** 3
**Empirical Novelty And Significance:** 3
**Recommendation:** 8
**Confidence:** 3

**Main Review:**

Strengths:

This paper provides a principled way to study the effect of example selection on convergence of SGD. By connecting the average stochastic gradient error to the SGD convergence, it reduces the SGD convergence problem to the problem of controlling the average stochastic gradient error. The latter problem is clearly much easier to address.

Compared to most prior works which usually focus on analyzing specific sampling/scanning strategies, this principled approach in this paper can potentially provide much more guidance on how to design new methods, and in which directions new approaches should go.

On the analysis of a few existing example orderings, the theory established in this paper can recover the convergence rates obtained before. This indicates that the convergence rate obtained through the analysis of average stochastic gradient error is tight enough, and should be useful for analyzing new sampling strategies in the future.

Weakness:

The proposed QMC-based algorithm is an extension of random reshuffling to the scenario of data augmentation. This is not building a new sampling strategy, but extending a known method to a different setting, which seems orthogonal to the main discussions of the paper.

The theory established in Section 3 has not touched “the deepest” yet. The assumption 2, which is the condition on average stochastic gradient error, remains a mathematical condition. So far, it is not quite clear how one should build an optimal sampling method, and how one can build a better sampling method than current ones, to accelerate SGD.


**Summary Of The Paper:**

1: This paper theoretically builds the following connection: if the averages of consecutive stochastic gradients converge faster to the full gradient, then the SGD with the corresponding sampling strategy will have a faster convergence rate.

2: Applying this theoretical finding on a few commonly used sampling strategies, the convergence rates established by prior works can be recovered.

3: Two algorithms are proposed based on this theory.


**Summary Of The Review:**

This paper provides a principled way to study the effect of example selection/sampling on convergence of SGD

---

> ### Author Response · Authors · 2021-11-15
> **Response to Reviewer rjuV**
>
> Thank you for the positive feedback, and we also hope that this general framework can simplify the analysis of any new SGD scan orders. As for your concerns, we now try to clarify them with some additional discussions.
>
> > 1. *The proposed QMC-based algorithm is an extension of random reshuffling to the scenario of data augmentation. This is not building a new sampling strategy, but extending a known method to a different setting, which seems orthogonal to the main discussions of the paper.*
>
> The QMC-based algorithm is actually an important part of sample ordering -- while other strategies like random reshuffling focus on ordering the empirical dataset, data augmentation can be seen as sampling over the population distribution. These two are correlated, as QMC-based sampling generally provides a similar error bound (Assumption 2 vs. Assumption 5), which allows us to obtain a better sequence of examples from the target distribution. The combination of random reshuffling with QMC-based data augmentation is two-fold: we first augment the data with QMC-based sampler, and then perform another sampler random reshuffling on these examples, so that we benefit from both sides.
>
> > 2. *The theory established in Section 3 has not touched “the deepest” yet. The assumption 2, which is the condition on average stochastic gradient error, remains a mathematical condition. So far, it is not quite clear how one should build an optimal sampling method, and how one can build a better sampling method than current ones, to accelerate SGD.*
>
> While it is true that for most applications it would be prohibitive to minimize
> the average gradient error in Assumption 2,
> our first attempt at doing so is given by the proposed greedy example ordering algorithm
> (Algorithm 1). One way to improve upon this is to develop faster algorithms to
> reorder a set of vectors that sum to one and each with bounded norm,
> with the goal of minimizing the sum of the initial partial sequences.
> Such algorithms may still be suboptimal but could significantly improve
> the runtime complexity of greedily minimizing the average gradient errors,
> and we leave this direction as future work.

---

### Decision · Program_Chairs · 2022-01-20

**Decision:**

Accept (Spotlight)

**Comment:**

This paper studies the dependency of SGD convergence on order of examples. The main observation of the paper is: if the averages of consecutive stochastic gradients converge faster to the full gradient, then the SGD with the corresponding sampling strategy will have a faster convergence rate. For different sampling strategies, sampling with replacement has slower convergence in stochastic gradient, where sampling without replacement can converge faster. The paper also proposes two new algorithms that can improve convergence rates in some interesting settings. The reviewers find the analysis clean and the new algorithms are interesting. There is some concern on the dependency on n or d for the faster rate, which should be discussed more clearly in the final version of the paper. Overall this is a solid contribution to the example selection problem.